# Structured Spectral Reasoning for Frequency-Adaptive Multimodal Recommendation

**Wei Yang**[1*]   **Rui Zhong**[1*]   **Yiqun Chen**[2*]   **Chi Lu**[1†]   **Peng Jiang**[1]
[1]Kuaishou Technology   [2]Renmin University of China
{yangwei08,luchi}@kuaishou.com,chenyiqun990321@ruc.edu.cn

## Abstract

Multimodal recommendation aims to integrate collaborative signals with heterogeneous content such as visual and textual information, but remains challenged by modality-specific noise, semantic inconsistency, and unstable propagation over user–item graphs. These issues are often exacerbated by naive fusion or shallow modeling strategies, leading to degraded generalization and poor robustness. While recent work has explored the frequency domain as a lens to separate stable from noisy signals, most methods rely on static filtering or reweighting, lacking the ability to reason over spectral structure or adapt to modality-specific reliability. To address these challenges, we propose a Structured Spectral Reasoning (**SSR**) framework for frequency-aware multimodal recommendation. Our method follows a four-stage pipeline: **(i) Decompose** graph-based multimodal signals into spectral bands via graph-guided transformations to isolate semantic granularity; **(ii) Modulate** band-level reliability with spectral band masking, a training-time masking with representation-consistency objective that suppresses brittle frequency components; **(iii) Fuse** complementary frequency cues using hyperspectral reasoning with low-rank cross-band interaction; and **(iv) Align** modality-specific spectral features via contrastive regularization to promote semantic and structural consistency. Experiments on three real-world benchmarks show consistent gains over strong baselines, particularly under sparse and cold-start settings. Additional analyses indicate that structured spectral modeling improves robustness and provides clearer diagnostics of how different bands contribute to performance. The code is available at https://github.com/llm-ml/SSR.git.

## 1 Introduction

Multimodal recommendation [1, 2, 3, 4] has become a central paradigm in modern information systems. It leverages auxiliary signals such as visual content, textual metadata and structural interactions to improve user preference modeling and content understanding [5, 6, 7, 8, 9]. These signals provide complementary semantics but also introduce modality-specific noise [10, 11, 12], semantic misalignment [13, 14] and representational redundancy [15, 16, 17]. The difficulties become more pronounced in graph-based architectures where user–item relations are propagated through neighborhoods [18, 19, 20]. In this setting, spurious cues from particular modalities, for example stylistic image patterns or ambiguous wording, can diffuse across the graph and affect otherwise unrelated nodes [21, 22, 23, 24]. The result is unstable representations and degraded performance, especially under cold-start or noisy conditions. Many existing approaches fuse modalities in the raw feature space [25, 26, 27], which limits control over how different semantic granularities interact across the graph. These challenges call for a structured approach to semantic disentanglement that isolates stable signals while suppressing unstable ones [28, 29, 30, 31].

---

[*]Equal contribution.
[†]Corresponding author.

39th Conference on Neural Information Processing Systems (NeurIPS 2025).

Recently, frequency-domain modeling has gained traction as a promising perspective [32, 33, 34]. By transforming signals into spectral components, models can expose structural patterns across different frequency bands [35, 36, 37]. Low-frequency components often encode smooth, global semantics such as long-term user intent or item popularity, while high-frequency components capture fine-grained variations or local inconsistencies [38, 39, 40]. This decomposition enables models to analyze and potentially filter modality-specific signals with greater precision. However, advanced approaches such as SMORE [15] and ChebyCF [41] largely rely on static reweighting or low-pass filtering and therefore lack explicit reasoning about band semantics or cross-frequency dependencies. Moreover, the use of frequency decomposition itself is often heuristic, without a principled mechanism to separate informative bands from noisy ones. Consequently, useful detail may be suppressed and unstable components may be retained, particularly when semantics vary across items and users.

These limitations are particularly acute in multimodal graph recommendation, where heterogeneous modalities interact within a topologically entangled space [42, 43, 44, 45]. Visual and textual semantics differ in granularity and abstraction, and they also differ in spectral distribution, which makes naïve fusion or static attention insufficient [15, 46, 47]. In this setting, robust representations should support adaptive modulation across frequency bands so that components influence one another based on semantic content and contextual relevance [38, 40, 48]. They should also enforce band-wise consistency across modalities to prevent collapse or redundancy within the decomposed space [49, 50]. However, current architectures lack a unified mechanism to model these structured relationships across frequency, modality and graph topology. At the same time, any solution must remain computationally tractable on large graphs and compatible with standard training pipelines. It should also provide clear diagnostics at the band level so that practitioners can inspect which frequencies and modalities drive recommendations. This motivates the need for approaches that account for band-level structure and maintain robustness across modalities.

To address these challenges, we propose **S**tructured **S**pectral **R**easoning (**SSR**) for frequency-aware multimodal graph recommendation. SSR follows a four-stage pipeline that *decomposes*, *modulates*, *fuses* and *aligns* signals within a shared spectral coordinate system. Motivated by the observation that frequency mapping reveals band-wise structure capturing collaborative semantics and modality-specific detail, we organize learning around band-level operations and supervision. Instead of treating bands as static or lightly reweighted features, SSR converts inputs into spectral representations and then reasons about their informativeness and stability. We introduce Spectral Band Masking (SBM), a training-time band-level perturbation with a representation-consistency objective that reduces reliance on brittle bands and improves robustness. We also design a graph-compatible hyperspectral operator (G-HSNO) that models cross-band and cross-modality dependencies through a compact low-rank parameterization. In addition, a spectral contrastive objective promotes band-wise cross-modal consistency without inference overhead. Together, these components provide a cohesive and efficient approach to structured spectral modeling for multimodal graphs.

In summary, the main contributions of this work are as follows:

- We present a unified four-stage framework for frequency-aware multimodal graph recommendation that places multimodal signals in a shared spectral space and performs band-wise decomposition, modulation, fusion, and alignment.

- We design a graph-compatible hyperspectral operator (G-HSNO) that models cross-band and cross-modality dependencies with a compact low-rank parameterization, achieving a strong accuracy and efficiency trade-off.

- Extensive experiments across multiple benchmarks show consistent gains over advanced baselines, with particularly strong improvements for cold-start users. Ablation and diagnostic analyses validate the effectiveness of each module.

## 2 RELATED WORK

### 2.1 Multi-modal Recommendation

Multimodal recommendation integrates auxiliary content such as images, text and audio to strengthen user preference modeling, especially under sparse supervision [51, 52, 53]. Early work augments collaborative filtering with side content [54, 55, 56]. Subsequent methods build modality-aware graph architectures to propagate and fuse signals along user–item relations [57, 58, 59, 60, 61, 62].

To mitigate cross-modal misalignment and semantic noise, recent approaches employ contrastive learning [63, 64], regularization [65, 66] and intention-aware designs [67, 68]. AlignRec [47] enforces modality–ID consistency, while PromptMM [69] enables lightweight adaptation. Generative strategies, including diffusion models [70], adversarial alignment [71] and large language models [72, 73, 74, 75], further improve robustness under heterogeneity. In addition, empowered by the reasoning capacity of large language models [76, 77] and multi-agent systems [78, 79, 80], recent advances such as Rec-GPT4V [1] and UniMP [81] leverage large vision–language models to strengthen multimodal semantic understanding and improve reasoning over heterogeneous content.

## 2.2 Frequency-aware Representation Learning

Frequency-domain modeling has recently emerged as a promising alternative to time- or topology-domain learning in recommendation tasks [34, 35]. In sequential modeling, various approaches leverage spectral transformations to capture user intent dynamics and mitigate temporal noise [36, 38], while others propose frequency-aware contrastive objectives or neural filtering for periodic behavior tracking [82, 83]. CFIT4SRec [36] adopts frequency-based contrastive learning frameworks, leveraging Fourier transformations to construct robust augmented views and improve representation learning. FEARec [38] and HMFTRec [34] add spectral attention and multi-frequency modules to capture low and high frequency signals beyond augmentation. For graph-based recommendation, ChebyCF [41], SelfGNN [84], and SComGNN [85] reformulate GNN propagation as spectral filtering, improving expressivity and denoising capacity. In multimodal settings, SMORE [15] explicitly models frequency-specific modality fusion, while FREEDOM [66] and MMGCL [64] implicitly introduce spectral diversity through augmentation. Nonetheless, most of these methods either treat frequency bands as static views or lack explicit modeling of band reliability and inter-band interactions. We address these gaps by pursuing adaptive, band-aware integration that regulates sensitivity to brittle components and supports interpretable cross-modal fusion.

## 3 Proposed Method

The problem definition can be found in Appendix A. In this section, we present **SSR**, a frequency-aware multimodal recommendation framework illustrated in Figure 1.

### 3.1 Spectral Transformation of Multimodal Graph Signals

Multimodal recommendation integrates ID embeddings with visual and textual features to learn unified user-item representations. A central challenge is cross-modal heterogeneity, where modalities reside in incompatible spaces and capture semantics at different granularities. Conventional fusion methods such as concatenation or attention operate in the raw feature domain and often ignore the structural dependencies induced by user-item interactions. Viewing these representations as graph signals enables frequency-aware modeling, where low-frequency components capture global preference patterns and high-frequency components encode fine-grained, modality-specific variations. Motivated by this perspective, we transform multimodal features into a unified frequency domain to facilitate band-level alignment and more interpretable fusion, supporting adaptive selection of frequency components for downstream recommendation.

**Graph Representation and Spectral Decomposition.** Given a user–item bipartite graph $\mathcal{G} = (\mathcal{N}, \mathcal{E})$ with adjacency matrix $\mathbf{A}$ [86, 87], we use the symmetric normalized Laplacian $\mathbf{L} = \mathbf{I} - \mathbf{D}^{-1/2}\mathbf{A}\mathbf{D}^{-1/2}$, where $\mathbf{D}$ is the degree matrix. For each node $n \in \mathcal{N}$, the associated signal (e.g., an ID embedding or a multimodal feature) is $\mathbf{x}_n \in \mathbb{R}^d$. Stacking all nodes yields $\mathbf{X} \in \mathbb{R}^{|\mathcal{N}| \times d}$. We analyze $\mathbf{X}$ in the spectral domain via the eigendecomposition $\mathbf{L} = \mathbf{U}\mathbf{\Lambda}\mathbf{U}^\top$, where $\mathbf{U}$ contains orthonormal eigenvectors and $\mathbf{\Lambda}$ is diagonal with eigenvalues. The graph Fourier transform (GFT) is $\widehat{\mathbf{X}} = \mathbf{U}^\top \mathbf{X}$ and the inverse transform is $\mathbf{X} = \mathbf{U}\widehat{\mathbf{X}}$. Each row of $\widehat{\mathbf{X}}$ corresponds to a graph frequency (eigenmode). In the next step we group these modes into bands for downstream processing. In practice, explicitly computing $\mathbf{U}$ is only feasible for small graphs. We thus adopt a hybrid implementation: for graphs with $|\mathcal{N}| \leq N_0$ we compute the full eigendecomposition of $\mathbf{L}$, while for larger graphs we avoid eigen-decomposition and approximate multi-band filtering using Chebyshev polynomials of the scaled Laplacian. In practice, we leverage structural signals from the interaction graph as well as auxiliary user–user and item–item relations to enrich the spectral pipeline.

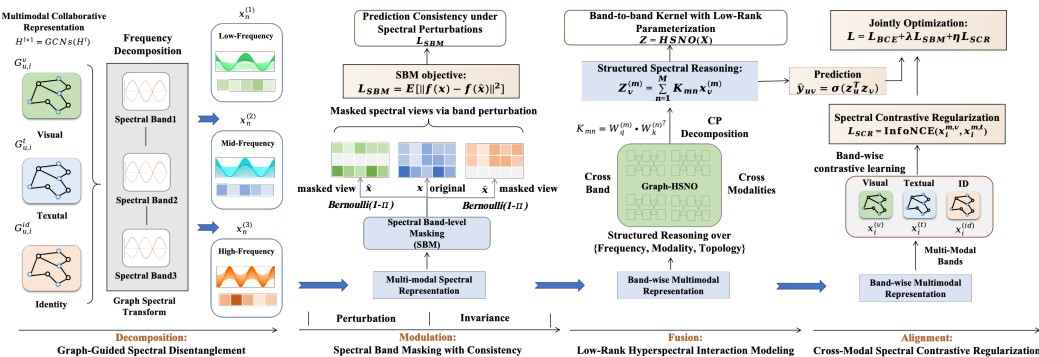

Figure 1: Overall architecture of our proposed framework. The model follows a structured four-stage pipeline: **(i) Decomposition** performs a modality-specific spectral transform on graph signals to disentangle multi-frequency components; **(ii) Modulation** applies Spectral Band Masking (SBM) to perturb and down-weight unreliable bands in a task-adaptive manner; **(iii) Fusion** leverages a low-rank Graph HyperSpectral Neural Operator (G-HSNO) to reason over cross-band and cross-modal dependencies; and **(iv) Alignment** introduces Spectral Contrastive Regularization (SCR) to enforce semantic consistency and spectral robustness across modalities.

**Frequency Band Construction.** We divide the spectrum into $M$ frequency bands using the energy distribution of spectral coefficients. Let $\widehat{\mathbf{x}}_k$ denote the $k$-th row of $\widehat{\mathbf{X}}$, and define its energy as $E_k = \|\widehat{\mathbf{x}}_k\|_2^2$. We compute the cumulative energy over ordered eigenmodes and partition the frequency axis into $M$ contiguous sets $\{\mathcal{F}_1, \ldots, \mathcal{F}_M\}$ such that each band contributes approximately equal total energy,

$$\sum_{k \in \mathcal{F}_m} E_k \approx \frac{1}{M} \sum_{k=1}^{|\mathcal{N}|} E_k, \tag{1}$$

which yields bands with comparable signal mass and avoids extreme imbalance across modes. Given the orthonormal basis $\mathbf{U} = [\mathbf{u}_1, \ldots, \mathbf{u}_{|\mathcal{N}|}]$, the reconstruction for band $m$ is $\mathbf{X}^{(m)} = \sum_{k \in \mathcal{F}_m} \mathbf{u}_k \widehat{\mathbf{x}}_k^\top$, where $\mathbf{u}_k$ is the $k$-th eigenvector. Each $\mathbf{X}^{(m)}$ captures information at a specific semantic resolution, with lower-index bands concentrating smoother components and higher-index bands emphasizing localized variations. In practice, $M$ is treated as a hyperparameter. When full eigenmodes are available, we partition bands by equal-energy quantiles of the cumulative spectrum. Otherwise, we split the spectrum in the eigenvalue domain and instantiate an $M$-band Chebyshev polynomial filter bank, which provides a scalable approximation to explicit band partitioning without eigendecomposition.

**Semantic-Aware Frequency Fusion.** Given the band-specific components $\{\mathbf{X}^{(m)}\}_{m=1}^M$, we aggregate them into a unified node representation with adaptive gates. Let $\mathbf{x}_n^{(m)}$ denote the $n$-th row of $\mathbf{X}^{(m)}$. The fused embedding for node $n$ is

$$\mathbf{z}_n = \sum_{m=1}^M \alpha_n^{(m)} \mathbf{x}_n^{(m)}, \tag{2}$$

where $\alpha_n^{(m)} \in \mathbb{R}_{\geq 0}$ are band-specific weights produced from node information. This allows each node to emphasize informative bands while down-weighting unstable ones. In practice, this aggregation can be implemented either by a lightweight gating module as above or by our subsequent G-HSNO operator, providing a unified and flexible interface for band fusion.

### 3.2 Spectral Band Masking with Consistency Training

Multimodal user–item graphs exhibit frequency components with different semantic roles and unequal reliability [15, 38]. Low-frequency bands tend to reflect smooth collaborative structure such as long-term preferences or popularity, whereas higher bands capture localized variation that is often modality sensitive. In practice, the latter can correlate with interactions yet fluctuate across items, users and contexts due to sampling noise, content artifacts and instance-specific patterns. Standard training

objectives amplify correlations that are easiest to exploit, which biases the predictor toward narrow spectral slices and encourages shortcut behavior. The result is increased sensitivity to small spectral perturbations and weaker generalization under sparsity or cold start.

To mitigate this issue, we introduce *Spectral Band Masking* (SBM), a training-time band-level robustness mechanism. During training, SBM randomly masks a subset of frequency bands and imposes a representation-consistency objective between the full-spectrum and masked inputs. This structured perturbation reduces over-reliance on any single band, encourages the model to distribute evidence across more stable components and improves robustness under distribution shift. SBM is lightweight, architecture-agnostic and integrates seamlessly with the spectral pipeline.

**Band-wise Masking.** Let $\{\mathbf{X}^{(m)}\}_{m=1}^{M}$ denote the band-wise features and let $\mathbf{x}^{(m)}$ be the per-instance slice. During training, we sample a binary mask independently for each instance,

$$\gamma_m^{(i)} \sim \text{Bernoulli}(1 - \pi), \quad m = 1, \ldots, M, \tag{3}$$

and construct a masked spectral view by dropping entire frequency bands:

$$\tilde{\mathbf{x}}^{(i)} = \sum_{m=1}^{M} \gamma_m^{(i)} \mathbf{x}^{(m)}. \tag{4}$$

In practice, we apply the mask on the band tensor and optionally rescale by the keep probability to keep the expected magnitude stable. To avoid degenerate all-zero masks, we ensure at least one band is kept for each instance. Unless otherwise specified, we use uniform masking over bands.

**Consistency Objective.** To stabilize learning under spectral perturbations, we encourage the representation computed from the masked view to match that of the full-spectrum view:

$$\mathcal{L}_{\text{SBM}} = \mathbb{E}_{\gamma} \big\| h(\mathbf{x}) - h(\tilde{\mathbf{x}}) \big\|_2^2. \tag{5}$$

where $h(\cdot)$ denotes the spectral encoder that outputs representations. This objective acts as band-aware perturbation with a representation-consistency regularizer. It discourages reliance on any single narrow band, reduces co-adaptation between bands and modalities, and distributes evidence across more stable components. $\mathcal{L}_{\text{SBM}}$ is optimized jointly with the main task loss. The objective adds negligible overhead and makes no changes at inference time.

### 3.3 Graph HyperSpectral Neural Operator (G-HSNO)

While frequency decomposition disentangles semantics across spectral bands, and band-level masking improves robustness to brittle cues, the system still needs a dedicated module to reason over frequency-structured inputs and extract task-relevant information. Traditional graph neural networks (e.g., GCNs) operate primarily in the spatial domain and often aggregate neighbor information uniformly or with simple attention. Such models are limited in capturing complex cross-frequency and cross-modality interactions, especially in multimodal recommendation where frequency semantics vary across modalities. A suitable module should explicitly model interactions among bands and modalities while respecting graph structure. It should also provide controllable capacity and low overhead so that it remains practical at scale.

To address this gap, we introduce the *Graph HyperSpectral Neural Operator (G-HSNO)*, a spectral operator that learns joint representations across three axes: frequency bands, modality types and graph topology. G-HSNO draws inspiration from neural operators in scientific computing and adapts the idea to discrete graph signals with multimodal frequency structure.

**Operator Design.** Let $\mathbf{x}_v^{(m)} \in \mathbb{R}^d$ denote the $m$-th frequency-band representation of node $v$. We aggregate all bands into a tensor $\mathbf{X} \in \mathbb{R}^{|\mathcal{N}| \times M \times d}$ and define G-HSNO as a spectral operator that models inter-band interactions. For each node $v$ and output band $m$,

$$\mathbf{Z} = \text{HSNO}(\mathbf{X}), \quad \mathbf{Z}_v^{(m)} = \sum_{n=1}^{M} \mathbf{K}_{mn} \mathbf{x}_v^{(n)}. \tag{6}$$

Collecting $\mathbf{Z}_v^{(m)}$ over all nodes yields $\mathbf{Z} \in \mathbb{R}^{|\mathcal{N}| \times M \times d}$. To reduce parameter complexity and avoid overfitting, we parameterize the kernel in a lightweight form. In particular, we adopt a diagonal

approximation $\mathbf{K}_{mn} = \mathrm{Diag}(\mathbf{k}_{mn})$, which has been widely used as an efficient spectral filtering variant. Concretely, we factorize $\mathbf{k}_{mn}$ into an inter-band mixing weight and a rank-wise diagonal filter: we compute a band-mixing score $s_{mn} = \langle \mathbf{W}_q(m,:), \mathbf{W}_k(n,:)\rangle/(\sqrt{r}\tau)$ and normalize it with a softmax function over output bands to $a_{mn}$, and we construct the diagonal filter as $\mathbf{k}_{mn} = a_{mn} \sum_{i=1}^{r} \rho_i \, \mathbf{v}^{(i)}$, where $\{\mathbf{v}^{(i)}\}_{i=1}^{r} \subset \mathbb{R}^d$ are rank-specific diagonal bases, and $\rho$ is either a global softmax weight or softly adapted by an auxiliary embedding to improve flexibility without introducing a full kernel.

**Graph-aware Enhancement.** To incorporate instance-specific information into spectral fusion, we learn a lightweight band reweighting function that conditions on the current node representation and its spectral components. Concretely, for node $v$, we compute a band-wise weight vector $\mathbf{g}_{\mathbf{v}}^{(\mathbf{m})}$. We then apply the band-wise reweighting to the mixed spectral tensor and aggregate over bands,

$$\mathbf{H}_v^{(m)} = \mathbf{g}_v^{(m)} \, \mathbf{Z}_v^{(m)}, \qquad \mathbf{z}_v = \sum_{m=1}^{M} \mathbf{H}_v^{(m)}. \tag{7}$$

To avoid undesired scale drift introduced by reweighting, we further apply a per-instance normalization that aligns the magnitude of the reweighted spectrum with the original one. Compared with fixed band aggregation, this step enables context-adaptive spectral fusion while keeping the overall representation scale stable.

## 3.4 Spectral Contrastive Regularization

To strengthen semantic alignment and structural integrity in the spectral space, we introduce a lightweight auxiliary objective called *Spectral Contrastive Regularization (SCR)*. Although G-HSNO models frequency-aware graph signals in a structured manner, representations from different modalities can still be inconsistent or collapse within the same band because of noise or weak alignment. Intuitively, components that lie in the same band (for example, image and text embeddings at band $m$) should encode comparable global or local semantics. In practice, distributional heterogeneity can cause divergence or entanglement with unrelated spectral patterns. SCR addresses this by applying a contrastive loss that encourages intra-band cross-modal consistency while preserving separation across bands. Optionally, the same intra-band contrastive regularization can also be applied to user-side spectral components to further stabilize the spectral space. Formally, let $\mathbf{x}_v^{(m,\mathrm{img})}$ and $\mathbf{x}_v^{(m,\mathrm{txt})}$ denote the $m$-th band representations of item $v$ from image and text. These form a positive pair. Negatives are sampled from either different items $v' \neq v$ at the same band or from any item at different bands $m' \neq m$. The loss is

$$\mathcal{L}_{\mathrm{SCR}} = -\log \frac{\exp\left(\mathrm{sim}(\mathbf{x}_v^{(m,\mathrm{img})}, \mathbf{x}_v^{(m,\mathrm{txt})})/\tau\right)}{\sum\limits_{(v',m')\neq(v,m)} \exp\left(\mathrm{sim}(\mathbf{x}_v^{(m,\mathrm{img})}, \mathbf{x}_{v'}^{(m',\cdot)})/\tau\right)}, \tag{8}$$

where $\mathrm{sim}(\cdot,\cdot)$ is cosine similarity, $\tau > 0$ is a temperature and $\mathbf{x}_{v'}^{(m',\cdot)}$ denotes a representation from either modality. This regularization aligns cross-modal signals within each band and enforces band-level discriminability. It introduces no extra inference cost and serves as a modular complement to the main objective. Our model is designed to be lightweight and efficient, with all frequency-aware modules implemented via sparse and batched operations.

## 3.5 Training and Optimization

The model is trained to predict user–item interaction probabilities from spectral-aware representations. For each pair $(u, v)$, the interaction score is $\hat{y}_{uv} = \sigma(\mathbf{z}_u^\top \mathbf{z}_v)$, where $\mathbf{z}_u$ and $\mathbf{z}_v$ are the final user and item embeddings, and $\sigma(\cdot)$ is the sigmoid function. The objective combines binary cross-entropy with above auxiliary losses: a band-masking consistency loss $\mathcal{L}_{\mathrm{SBM}}$ and a spectral contrastive loss $\mathcal{L}_{\mathrm{SCR}}$ that aligns modality-specific features within each band. Further, we introduce a simple squared regularization on the HSNO kernel and filter to improve training stability. The final loss is

$$\mathcal{L} = \mathcal{L}_{\mathrm{BCE}} + \lambda \, \mathcal{L}_{\mathrm{SBM}} + \eta \, \mathcal{L}_{\mathrm{SCR}} + \gamma \, \mathcal{L}_{\mathrm{Reg}}, \tag{9}$$

where $\lambda, \eta, \gamma \geq 0$ control the strength of the auxiliary terms. We optimize the model end to end with mini-batch stochastic gradient descent. No masking or rescaling is applied at inference time.

# 4 Experiments

## 4.1 Experimental Settings

**Datasets and Baselines.** We conduct experiments on three standard Amazon subsets (Baby, Sports, Clothing) using full-ranking metrics (Recall@K, NDCG@K). A broad range of collaborative, multimodal, and graph-based baselines are compared. For full dataset details, preprocessing, and baseline list, please refer to Appendix B.

## 4.2 Overall Performance Comparison (RQ1)

Table 1 summarizes the top-K recommendation performance across three multimodal Amazon datasets. Our proposed SSR achieves consistent and significant improvements over all baseline models. This performance advantage is particularly evident on sparse datasets like Clothing, where SSR outperforms prior frequency-aware models such as SMORE by +5.0% in Recall@10. These gains can be attributed to SSR's ability to capture fine-grained, frequency-aware multimodal semantics that are often overlooked or insufficiently captured in prior approaches. While recent methods like SMORE leverage frequency-domain projection to suppress modality noise, they treat frequency components uniformly and lack structural modeling of collaborative signals. AlignRec and MMIL introduce advanced alignment or intention modeling mechanisms, but still operate in the time domain and fail to differentiate spectral semantics at varying granularity. In contrast, SSR explicitly decomposes multimodal representations into interpretable frequency bands via spectral graph transformation, dynamically fuses them through gated hyper-spectral operations, and further regularizes mid-band interactions via spectrum-aware contrastive learning. This design enables SSR to not only denoise but also extract highly discriminative and complementary signals across modalities and frequency levels.

Table 1: Recommendation performance on three Amazon benchmarks. SSR achieves consistent improvements over collaborative, graph-based, and multi-modal baseline.

| Dataset | Metric | BPR | LightGCN | VBPR | MMGCN | GRCN | DualGNN | RecVAE | LATTICE | BM3 | FREEDOM | DiffMM | MMIL | AlignRec | SMORE | SSR |
|---|---|---|---|---|---|---|---|---|---|---|---|---|---|---|---|---|
| Baby | R@10 | 0.0357 | 0.0479 | 0.0418 | 0.0413 | 0.0538 | 0.0507 | 0.0501 | 0.0561 | 0.0573 | 0.0624 | 0.0617 | 0.0670 | 0.0674 | 0.0680 | **0.0728** |
| | R@20 | 0.0575 | 0.0754 | 0.0667 | 0.0649 | 0.0832 | 0.0782 | 0.0811 | 0.0867 | 0.0904 | 0.0985 | 0.0978 | 0.1035 | 0.1046 | 0.1035 | **0.1103** |
| | N@10 | 0.0192 | 0.0257 | 0.0223 | 0.0211 | 0.0285 | 0.0264 | 0.0275 | 0.0305 | 0.0311 | 0.0324 | 0.0321 | 0.0361 | 0.0363 | 0.0365 | **0.0395** |
| | N@20 | 0.0249 | 0.0328 | 0.0287 | 0.0275 | 0.0364 | 0.0335 | 0.0358 | 0.0383 | 0.0395 | 0.0416 | 0.0408 | 0.0455 | 0.0458 | 0.0457 | **0.0491** |
| Sports | R@10 | 0.0432 | 0.0569 | 0.0561 | 0.0394 | 0.0607 | 0.0574 | 0.0603 | 0.0628 | 0.0659 | 0.0705 | 0.0687 | 0.0747 | 0.0758 | 0.0762 | **0.0825** |
| | R@20 | 0.0653 | 0.0864 | 0.0857 | 0.0625 | 0.0928 | 0.0881 | 0.0916 | 0.0961 | 0.0987 | 0.1077 | 0.1035 | 0.1133 | 0.1160 | 0.1142 | **0.1203** |
| | N@10 | 0.0241 | 0.0311 | 0.0305 | 0.0203 | 0.0335 | 0.0316 | 0.0348 | 0.0339 | 0.0357 | 0.0382 | 0.0357 | 0.0405 | 0.0414 | 0.0408 | **0.0449** |
| | N@20 | 0.0298 | 0.0387 | 0.0386 | 0.0266 | 0.0421 | 0.0393 | 0.0431 | 0.0431 | 0.0443 | 0.0478 | 0.0458 | 0.0505 | 0.0517 | 0.0506 | **0.0547** |
| Clothing | R@10 | 0.0206 | 0.0361 | 0.0283 | 0.0221 | 0.0425 | 0.0447 | 0.0330 | 0.0503 | 0.0577 | 0.0616 | 0.0593 | 0.0643 | 0.0651 | 0.0659 | **0.0708** |
| | R@20 | 0.0303 | 0.0544 | 0.0418 | 0.0357 | 0.0661 | 0.0663 | 0.0485 | 0.0755 | 0.0845 | 0.0917 | 0.0874 | 0.0961 | 0.0993 | 0.0987 | **0.1032** |
| | N@10 | 0.0114 | 0.0197 | 0.0162 | 0.0116 | 0.0227 | 0.0237 | 0.0187 | 0.0277 | 0.0316 | 0.0333 | 0.0325 | 0.0348 | 0.0356 | 0.0360 | **0.0386** |
| | N@20 | 0.0138 | 0.0243 | 0.0196 | 0.0151 | 0.0283 | 0.0289 | 0.0227 | 0.0356 | 0.0387 | 0.0409 | 0.0396 | 0.0428 | 0.0437 | 0.0443 | **0.0466** |

## 4.3 Cold-start User Performance (RQ2)

To evaluate model robustness under user cold-start scenarios, we examine top-K recommendation performance for users with limited interaction history. As shown in Table 2, our proposed SSR consistently outperforms all baselines across three datasets and four evaluation metrics, with particularly strong gains in Recall@20 metrics (e.g., +10.0% over SMORE on *Baby*). These improvements validate our key design principle: rather than assuming a uniform representation fusion for all users, SSR leverages frequency-aware decomposition and user-specific gating to tailor the semantic emphasis across low, mid, and high bands. Compared to prior models such as SMORE and MMIL, which adopt static fusion schemes or modality-aligned latent spaces, SSR introduces gated hyperspectral operations and spectral contrastive regularization that adaptively enhance discriminative cues even under sparse data. The model's ability to selectively amplify high-frequency discriminative signals is particularly important for personalizing recommendations when strong interaction priors are unavailable, making it especially valuable in cold-start settings. Moreover, the consistent lead across all frequency bands highlights the efficacy of our frequency decomposition design in isolating modality-relevant semantics and suppressing irrelevant noise. These results suggest that SSR not only performs well under typical conditions but also generalizes effectively to sparse and unseen user behaviors, confirming its value as a robust and adaptive recommendation framework.

Table 2: Performance comparison in cold-start user scenarios (users with $\leq 5$ interactions). SSR consistently outperforms strong baselines, demonstrating its robustness in sparse conditions and effectiveness in adaptive multimodal modeling.

| Model | Amazon-Baby | | | | Amazon-Sports | | | | Amazon-Clothing | | | |
|---|---|---|---|---|---|---|---|---|---|---|---|---|
| | R@10 | R@20 | N@10 | N@20 | R@10 | R@20 | N@10 | N@20 | R@10 | R@20 | N@10 | N@20 |
| GRCN | 0.0510 | 0.0770 | 0.0268 | 0.0333 | 0.0556 | 0.0814 | 0.0305 | 0.0370 | 0.0416 | 0.0648 | 0.0213 | 0.0271 |
| BM3 | 0.0549 | 0.0845 | 0.0288 | 0.0363 | 0.0581 | 0.0920 | 0.0305 | 0.0390 | 0.0424 | 0.0614 | 0.0229 | 0.0276 |
| LATTICE | 0.0570 | 0.0827 | 0.0298 | 0.0362 | 0.0380 | 0.0570 | 0.0205 | 0.0252 | 0.0414 | 0.0555 | 0.0218 | 0.0254 |
| FREEDOM | 0.0535 | 0.0880 | 0.0297 | 0.0384 | 0.0622 | 0.0933 | 0.0328 | 0.0406 | 0.0444 | 0.0671 | 0.0241 | 0.0298 |
| MMIL | 0.0672 | 0.1014 | 0.0377 | 0.0463 | 0.0759 | 0.1141 | 0.0422 | 0.0517 | 0.0635 | 0.0948 | 0.0341 | 0.0423 |
| SMORE | 0.0687 | 0.1019 | 0.0381 | 0.0464 | 0.0788 | 0.1158 | 0.0420 | 0.0512 | 0.0676 | 0.0983 | 0.0369 | 0.0446 |
| SSR | **0.0765** | **0.1125** | **0.0424** | **0.0516** | **0.0832** | **0.1227** | **0.0439** | **0.0541** | **0.0698** | **0.1012** | **0.0387** | **0.0467** |

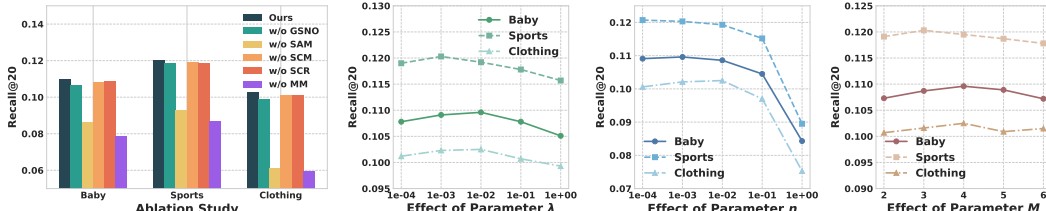

Figure 2: Ablation and sensitivity analysis. The left plot shows the impact of removing key components from SSR, validating the effectiveness of each module. The right three plots illustrate the influence of the information bottleneck weight $\lambda$, contrastive loss weight $\eta$, and the number of frequency bands $M$, confirming the stability of SSR under a range of hyperparameter settings.

## 4.4 Ablation Study and Sensitivity Analysis (RQ3)

We conduct ablations to quantify the contribution of each component: semantic-aware fusion (SAF), the hyperspectral operator (G-HSNO), spectral regularizers (SBM and SCR), and multimodal inputs. As shown in Fig. 2, removing any component leads to clear performance degradation. Notably, removing **SAM** causes the most significant drop, confirming the importance of semantically guided decomposition. Excluding **G-HSNO** also harms accuracy, indicating that cross-band and cross-modality reasoning is vital for capturing diverse user–item patterns. **SBM** and **SCR** yield smaller but consistent gains, improving robustness and alignment. Eliminating multimodal inputs and using ID features only leads to the weakest results, highlighting the utility of complementary content signals. These findings show that spectral semantics, adaptive fusion and modality reasoning act in a complementary manner within our architecture. We also perform a comprehensive sensitivity analysis over $\lambda$, $\eta$, and $M$, and observe consistently stable performance across a broad and practically relevant range of hyperparameter settings (the detailed results are reported in Appendix C.1).

## 4.5 Cross-Modality Distribution in Frequency Bands (RQ4)

To investigate how frequency decomposition shapes cross-modal representations, we visualize the distribution of user embeddings across low, mid and high bands for three modalities: ID, visual and textual. As shown in Fig. 3, in the low-frequency band, the three modalities form distinct and well-separated clusters, indicating that this frequency component preserves modality-specific semantics such as identity priors, visual styles, and coarse-grained textual categories. In the mid-frequency band, modality boundaries begin to blur, revealing interaction patterns and semantic alignment across modalities. This observation is consistent with applying frequency-aware fusion using G-HSNO and band-level regularization using SBM to capture synergistic cross-modal signals. In the high-frequency band, the embeddings from different modalities become densely entangled, suggesting the emergence of fine-grained, modality-agnostic representations that are highly discriminative but also sensitive to noise. This validates the necessity of spectral decomposition as a core design: it not only disentangles semantics across frequency and modality axes but also provides an interpretable basis for adaptive fusion and regularization. The observed patterns strongly support our hypothesis that frequency-aware modeling enables structured representation evolution, and reinforce the design effectiveness of our SBM and G-HSNO components.

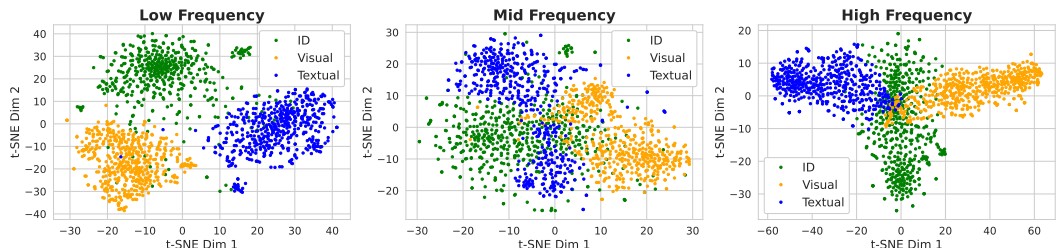

Figure 3: **t-SNE visualization of user embeddings across frequency bands.** Each subfigure illustrates ID, Visual, and Textual embedding distributions under a specific band.

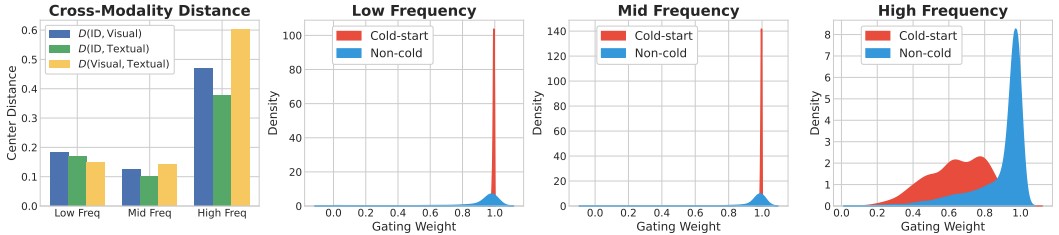

Figure 4: **Left**: Cross-modality center distances across frequency bands, highlighting increasing alignment in mid-frequency and modality-agnostic fusion in high-frequency regions. **Right**: KDE plots of frequency gating weights for cold-start vs. non-cold users.

### 4.6 Cross-Frequency Analysis of Modality Center Distances (RQ5)

We further assess cross-modal alignment across frequency bands by measuring pairwise distances between the centers of the ID, visual and textual representations. Concretely, for each band we compute a modality center as the mean embedding over items and then evaluate distances between modality centers. As shown in Fig. 4, the distances are minimized in the mid-frequency region, indicating that this band best captures shared cross-modal semantics. In contrast, low-frequency bands encode stable, modality-specific structure, whereas high-frequency bands reflect sparse or noisy variation. These asymmetric patterns motivate differentiated modeling across bands. Our design leverages this by applying spectrum-specific fusion (G-HSNO) and band-level regularization (SBM) in the mid-frequency region, where cross-modal synergy is most pronounced.

### 4.7 Adaptive Spectral Modulation under Representation Uncertainty (RQ6)

Our design is grounded in the idea that frequency components reflect different semantic roles, and that adapting their influence based on input uncertainty is key to effective recommendation. To validate this hypothesis, we conduct an analysis under the cold-start scenario. As shown in Fig. 4, cold-start items rely more on low and mid frequency signals, while non-cold items emphasize high-frequency cues. This behavior emerges from the interplay of spectral decomposition, G-HSNO, and contrastive regularization. These findings highlight that spectral semantics are not uniformly informative, and their utility is tightly coupled with data sparsity and user context. This also suggests that frequency-aware mechanisms can serve as a proxy for uncertainty estimation in representation learning. A thorough and comprehensive analysis is provided in Appendix C.3.

## 5 Conclusion

In this paper, we propose a structured spectral learning framework for multimodal graph recommendation that addresses modality noise, semantic inconsistency, and propagation instability via a four-stage pipeline: decomposition, band-level modulation, hyperspectral fusion, and alignment. Extensive experiments validate its effectiveness and robustness across diverse datasets. Future directions include integrating LLMs for reasoning-aware spectral graphs, designing adaptive spectral controllers, and extending to broader graph inference tasks such as dynamic modeling and temporal reasoning.

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

# A  Problem Formulation

We focus on the task of personalized recommendation with multimodal item content, where the goal is to predict user preferences by leveraging both historical interactions and rich item descriptions. Let $\mathcal{U}$ and $\mathcal{V}$ denote the sets of users and items, respectively. The observed implicit feedback is represented by a binary interaction matrix $\mathbf{R} \in \{0,1\}^{|\mathcal{U}| \times |\mathcal{V}|}$, where $\mathbf{R}_{uv} = 1$ indicates that user $u$ has interacted with item $v$, and $\mathbf{R}_{uv} = 0$ denotes unobserved behavior. Each user $u \in \mathcal{U}$ and item $v \in \mathcal{V}$ is associated with an ID-based embedding $\mathbf{e}_u^{\mathrm{id}}, \mathbf{e}_v^{\mathrm{id}} \in \mathbb{R}^{d_{\mathrm{id}}}$ that encodes collaborative filtering signals. In addition, each item is accompanied by multimodal content from a set of modalities $\mathcal{C} = \{\mathrm{img}, \mathrm{txt}\}$, where $\mathbf{e}_v^{(c)} \in \mathbb{R}^{d_c}$ denotes the representation of item $v$ in modality $c \in \mathcal{C}$.

To unify both interaction and multimodal semantics, we formulate the recommendation scenario as a graph-based learning problem. Specifically, we construct a bipartite user-item graph $\mathcal{G} = (\mathcal{N}, \mathcal{E})$, where the node set $\mathcal{N} = \mathcal{U} \cup \mathcal{V}$ contains users and items, and the edge set $\mathcal{E} \subseteq \mathcal{U} \times \mathcal{V}$ reflects observed interactions. The adjacency matrix $\mathbf{A} \in \{0,1\}^{|\mathcal{N}| \times |\mathcal{N}|}$ captures the connectivity structure of the interaction graph.

Under this formulation, the ID embeddings and multimodal features can be treated as semantic signals defined over the graph $\mathcal{G}$. The recommendation task thus becomes a graph signal inference problem: learning a spectral-aware node representation function $f : \mathcal{N} \to \mathbb{R}^d$ that jointly captures topological context and content semantics. The predicted relevance score between user $u$ and item $v$ is computed as

$$\hat{r}_{uv} = f(u)^\top f(v), \tag{10}$$

where $f(\cdot)$ denotes the frequency-aware and task-adaptive representation for each node. The system then ranks all candidate items based on $\hat{r}_{uv}$ and recommends the top-$N$ most relevant items.

# B  Experimental Settings

## B.1  Datasets

To comprehensively evaluate the effectiveness of our proposed framework, we conduct experiments on three representative subsets of the Amazon Product Review corpus[3], widely adopted in multimodal recommendation literature [15, 51]. We select categories with diverse product types and rich multimodal metadata: *Baby*, *Sports and Outdoors*, and *Clothing, Shoes and Jewelry* (referred to as Baby, Sports, and Clothing). Each dataset contains explicit user-item interaction records along with high-quality product images and textual descriptions. To ensure data reliability and evaluation fairness, we apply the standard 5-core preprocessing, retaining users and items with at least five interactions. These features serve as the multimodal input for all methods that support side content. Dataset statistics are reported in Table 3.

Table 3: Statistics of the Amazon datasets used in our experiments.

| Dataset | #Users | #Items | #Interactions | Interaction Density |
|---|---|---|---|---|
| Baby | 19,445 | 7,050 | 139,110 | 0.101% |
| Sports | 35,598 | 18,357 | 256,308 | 0.039% |
| Clothing | 39,387 | 23,033 | 237,488 | 0.026% |

## B.2  Baseline Models

We benchmark our approach against a diverse suite of strong baseline models spanning three major categories: collaborative filtering (CF), multimodal recommendation (MM), and graph-based architectures (GNN-based). In the CF category, we consider **BPR** [88], a foundational pairwise ranking model based on matrix factorization, and **LightGCN** [89], which simplifies GCNs by removing non-linear components for efficient collaborative signal propagation. For multimodal baselines, we include **VBPR** [51], which enhances MF with visual signals; **MMGCN** [57], **GRCN** [58], and **Dual-GNN** [59], which incorporate modality-aware graph learning designs. We further compare with more

---

[3]http://jmcauley.ucsd.edu/data/amazon/

recent models that incorporate autoencoder, self-supervised learning or regularization techniques, such as **RecVAE** [90],**SLMRec** [62], **LATTICE** [65], **BM3** [63], and **FREEDOM** [61]. Finally, we include advanced models leveraging diffusion processes and alignment strategies, including **DiffMM** [70], **MMIL** [67], **AlignRec** [47], and the frequency-aware method **SMORE** [15].

### B.3 Evaluation Protocol

We evaluate with a full ranking protocol that reflects practical deployment. For each test user the model ranks all items not seen in training. We report **Recall@K** and **NDCG@K** for $K \in \{10, 20\}$. Recall@K measures top K hit coverage. NDCG@K measures rank aware relevance.

We build splits in chronological order with an 80/10/10 ratio for train, validation, and test. During training we use negative sampling. Each positive interaction is paired with one randomly chosen item that the user did not interact with to provide contrastive supervision.

### B.4 Implementation Details

We implement our method using PyTorch within the MMRec benchmarking framework [91]. We use 64-dimensional embedding vectors for all item and user representations and apply Xavier initialization [92]. The model is optimized using the Adam optimizer [93] with a batch size of 2048. Learning rates are tuned from $\{0.0001, 0.0005, 0.001, 0.005\}$ using the validation set. Early stopping is triggered if Recall@20 does not improve within 20 validation steps. We conduct extensive ablations on the number of frequency bands $M \in \{2, 3, 4, 5\}$ and evaluate regularization weights for the spectral consistency loss, contrastive term and regularization term from $\{0.0001, 0.001, 0.01, 0.1, 1.0\}$. All hyperparameter choices are selected based on validation performance. Our code and preprocessed datasets will be released to ensure reproducibility.

## C Experiments Analysis

### C.1 Sensitive Analysis

To evaluate the robustness and generalizability of our approach, we conduct parameter sensitivity experiments on three key hyperparameters: the spectral masking regularization weight $\lambda$, the spectral contrastive regularization weight $\eta$, and the number of frequency bands $M$ used in decomposition. As shown in Fig. 2, our model exhibits stable performance across a wide range of values, underscoring its robustness to hyperparameter choices. For $\lambda$, we observe a consistent performance peak around $10^{-2}$, with degradation when $\lambda$ becomes excessively large ($\geq 0.1$), suggesting that overly aggressive suppression of spectral noise may harm useful information flow. A similar pattern emerges for $\eta$, where moderate contrastive regularization improves spectral alignment, but large values quickly deteriorate performance due to over-constraining the representation space. Finally, varying the number of frequency bands $M$ reveals that a three- or four-band decomposition provides the best trade-off: too few bands limit spectral expressiveness, while too many introduce noise and overfitting. These findings validate our design principle of moderate, structure-aware spectral decomposition with targeted regularization, and demonstrate that our model achieves strong results without requiring extensive hyperparameter tuning.

### C.2 Cross-Frequency Analysis of Modality Center Distances

We examine the spectral dynamics of modality alignment by measuring pairwise distances between the centers of ID, visual and textual representations across frequency bands. As shown in Fig. 4, the distances are smallest in the mid frequency region. This suggests that this band captures shared semantic signals across modalities. The effect is meaningful and not due to chance. Cross modal coherence is not uniform across the spectrum. It peaks at intermediate spectral scales. The low frequency region encodes stable modality specific structure such as identity priors or global style. The high frequency region highlights sharp or noisy components that are often unique to one modality. These asymmetric patterns support differentiated strategies for each band. Our architecture follows this idea. It applies spectrum specific fusion (G-HSNO) and regularization (SBM) in the mid frequency band where cross modal synergy is strongest. This analysis supports our central claim.

Frequency aware modeling exposes the spectral locality of modality alignment and clarifies semantic granularity that conventional fusion would hide.

### C.3  Adaptive Spectral Modulation under Representation Uncertainty

Our framework is built on the premise that different frequency components carry distinct semantic roles, ranging from stable structural priors to fine-grained discriminative cues, and that effective recommendation requires instance-aware modulation across this spectrum. To validate this capability, we examine a controlled setting where item representations vary in semantic certainty, namely the cold-start scenario. As shown in Fig. 4, cold-start items exhibit dominant reliance on low- and mid-frequency signals, reflected by sharply peaked spectral weights in these bands, while high-frequency components are significantly attenuated. Non-cold items, in contrast, show more distributed and high-frequency-skewed preferences that leverage richer personalized cues. This divergence arises not from a single module but from our unified design that combines spectral decomposition, gated hybrid spectral operators (G-HSNO), and contrastive regularization. The model does not apply a static fusion; it learns to regulate spectral emphasis based on input uncertainty. This analysis empirically substantiates the necessity of frequency-aware modeling and confirms the system's ability to achieve fine-grained, context-adaptive representation control.

### C.4  Discussion

Our study demonstrates that frequency-aware modeling provides a structured and interpretable lens for understanding multimodal interactions in recommendation systems. By decomposing user–item representations into distinct spectral bands, the proposed framework captures complementary semantic cues across modalities and enables adaptive fusion guided by spectral semantics. The integration of band-level modulation and spectrum-level regularization further enhances robustness against modality noise and data sparsity, allowing the model to dynamically adjust its emphasis based on uncertainty and interaction context. This design not only improves performance but also offers a more transparent explanation of how multimodal signals contribute to preference formation across different frequency levels. Despite its effectiveness, our method still has several limitations. First, while spectral decomposition enhances interpretability, it introduces additional computational cost compared to purely time- or topology-domain approaches. Second, the selection of decomposition granularity (i.e., the number of frequency bands) remains heuristic and may require dataset-specific tuning. Third, although our model captures intra- and inter-band correlations, it assumes a fixed transformation basis that might not fully generalize to highly non-stationary multimodal patterns. Finally, the current framework focuses on static multimodal graphs, extending it to temporal or dynamic user behaviors represents an important future direction. Addressing these limitations could further strengthen the generality and scalability of frequency-aware recommendation systems.

