# OpenReview forum: "Structured Spectral Reasoning for Frequency-Adaptive Multimodal Recommendation"
_NeurIPS.cc/2025/Conference — NeurIPS 2025 poster_

### Official Review · Reviewer_KUNi · 2025-06-30

**Clarity:** 2
**Significance:** 3
**Originality:** 3
**Rating:** 5
**Confidence:** 4

**Summary:**

The paper proposes SSR, short for Structured Spectral Reasoning, a novel multimodal recommender system which works in the frequency domain and adaptively utilizes the graph and multimodal spectral signals to solve known issues in the multimodal recommendation literature.

Unlike similar previous approaches, SSR performs a four stages pipeline, involving: (i) the decomposition of graph and multimodal signals into spectral bands to obtain a more granular representation; (ii) a modulation of the signal's reliability with some random masking of the spectral bands, to make the model robust to spurious frequency components; (iii) the fusion of the decomposed spectral bands into one unique representation; (iv) the alignment of spectral bands and modalities through contrastive learning.

To begin with, the graph and multimodal spectral signals are decomposed into spectral bands, and for each node in the dataset, they are fused with a gated attention weight.

Second, a masking mechanism is applied to randomly perturb the spectral bands during the training and to make the model robust to spurious frequency components; this objective is obtained through invariant risk minimization, a loss component of the overall loss function where the normal and the counterfactual settings are supposed to be close in their learned representations.

Then, the model introduces a Graph HyperSpectral Neural Operator (G-HSNO) which can work three axes, namely, frequency bands, modality types, and graph topology. In this respect, the authors adopt Canonical Polyadic (CP) decomposition, a low-rank factorization of the kernel tensor adopted in the G-HSNO operator to reduce the overfitting and the parameter complexity.

Finally, a Spectral Contrastive Regularization (SCR) is introduced to perform a contrastive learning loss encouraging intra-band modality consistency but preserving, at the same time, inter-band separation.

Extensive experimental results on three popular datasets from the Amazon Reviews collection, against 14 state-of-the-art recommender and multimodal recommender systems demonstrate the efficacy of the proposed approach. Further analyses show the goodness of the approach also in cold-start scenarios. Then, an ablation study and a hyper-parameter sensitivity analyses prove the soundness of the architectural choices made by the authors. Finally, the authors show how effective is the framework at adapting at different decomposed frequencies across modalities, how well-aligned are the modalities (especially at the mid frequency), and how the adaptive spectral modulation can tailor different settings, such as cold- and non-cold-start items.

**Questions:**

My questions are mainly related to some of the weaknesses I pointed out above.

Q1) Do the authors think that the proposed framework would have suited better to other scenarios than multimodal recommendation? For instance, knowledge graphs information in recommendation? And, in case, why did they choose to test it with multimodal recommendation?

Q2) How different is the proposed approach from similar solutions which perform disentanglement in the spatial domain? What is the benefit of performing it in the frequency domain?

Q3) Why did the authors go for CP in the kernel tensor decomposition, and did they try other solutions before coming up with CP?

Q4) Why did the authors use gated attention in the semantic fusion of spectral bands? What are the specific benefits of it, and did they try with other similar solutions?

**Ethical Concerns:**

["NO or VERY MINOR ethics concerns only"]

**Final Justification:**

After the rebuttal phase, after reading the other reviews, and authors' responses to those and to the careful comments raised by the Area Chair, I decide to confirm my initial score I gave to the paper.

Specifically, all my concerns have been discussed and addressed by the authors, and (to the best of my understanding), also the concerns raised by the Area Chair have been discussed and addressed by the authors.

Whatever the final outcome for this paper, I think it will benefit quite a lot from the fruitful discussion we had along with the authors, the reviewers, and the Area Chair. I believe the paper will improve its quality a lot in any case.

**Limitations:**

Yes.

**Paper Formatting Concerns:**

I do not see any major formatting issues in this paper.

**Quality:**

3

**Strengths And Weaknesses:**

**Strengths**

$\bullet$ The paper lays in one of the current hot topics in the literature of multimodal recommendation, namely, the translation of the task of multimodal recommendation to the frequency domain from the common spatial domain.

$\bullet$ In relation to the first point, the authors clearly highlight the main limitations in current approaches from the literature (also recent ones which work in the spectral domain), and position the proposed approach within the state-of-the-art.

$\bullet$ The methodology seems adequately sound and novel, where all main stages of the pipeline are clearly reported in the formalism and also in Figure 1. All theorems are adequately supported by the proofs as reported in the Supplementary Material.

$\bullet$ The experimental setting is quite extensive, with several analyzed dimensions which allow not only to show the efficacy of the proposed approach under overall performance perspectives, but also under specific evaluation dimensions which justify the effectiveness of each idea conveyed in the paper.

**Weaknesses**

$\bullet$ To my understanding, the proposed framework could potentially be applied to many different scenarios where other side information is available apart from the simple collaborative signal. This is especially evident since the multimodal aspect appears only in Section 3.4 after the whole presentation of the methodology, which appears quite agnostic of the modalities. While this could indeed be a positive aspect of the approach (i.e., to be side-information agnostic), do the authors think that maybe the approach would have suited better to other scenarios? Something like knowledge graphs information in recommendation? And, in case, why did they choose to test it with multimodal recommendation?

$\bullet$ From my perspective, I see how the proposed approach has something in common with previous approaches in (multimodal) recommendation, such as DGCF [\*] and GRCN. Specifically, the two mentioned approaches work by disentangling the embedding representation of nodes (and modalities) into aspects, which are learned end-to-end in the downstream task. To me, this appears as a spatial version of the frequency disentanglement of the spectral bands. Can the authors elaborate a little bit on that? How effective is translating the whole setting to the spectral domain instead of keeping it limited to the spatial one?

$\bullet$ Regarding the kernel tensor decomposition, the literature outlines a wide range of possible solutions apart from CP. For instance, another possible technique is Tucker decomposition, which usually allows more flexibility than CP. Why did the authors go for CP, and did they try other solutions before coming up with CP? I think this aspect should be better clarified. A similar consideration could be done for the semantic-aware frequency fusion. Why did the authors use gated attention? What are the specific benefits of it, and did they try with other similar solutions? This aspect should also be better clarified.

$\bullet$ The code is not released at review time, thus not ensuring complete reproducibility of the approach. While I acknowledge the usage of the framework MMRec for all the used baselines, I think the work would benefit from sharing the code also of the proposed approach.

**References**

[\*] Xiang Wang, Hongye Jin, An Zhang, Xiangnan He, Tong Xu, Tat-Seng Chua: Disentangled Graph Collaborative Filtering. SIGIR 2020: 1001-1010

---

> ### Author Rebuttal · Authors · 2025-07-28
>
> We sincerely thank you for your thorough and insightful review, and appreciate the valuable comments and discussions you provided. We are especially grateful for your deep understanding of our work, it is clear that you have fully grasped the core ideas of our approach, and the questions you raised are highly meaningful. We have prepared the following responses in the hope of addressing some doubts.
>
> **Q1| About Application Scenarios**
>
> Our framework is designed to be general and extensible, accommodating various types of auxiliary signals by modeling them as graph-defined fields and analyzing their frequency characteristics. In that sense, choosing images‑and‑texts for the first study is an instantiation rather than a restriction. The same framework remains valid so long as two conditions hold:
> 1) the side information can be represented on the same (or an aligned) node set;
> 2) frequency carries meaning that the downstream task cares about.
>
> **(1) Knowledge‑graph Recommendation as a Natural Extension**
>
> Knowledge graphs (KGs) offer rich relational structures that naturally align with our spectral reasoning paradigm. Each relation induces its own Laplacian, whose spectral components reflect both hierarchical and fine-grained semantics, analogous to the frequency bands in multimodal signals.
>
> In this setting:
> - Our **spectral decomposition** generalizes to relation-wise spectra, enabling structured reasoning without relying on handcrafted metapaths. This allows the model to respect the inherent relational geometry of the KG while avoiding manual path enumeration or rigid schema constraints.
> - **Causal masking** offers a principled alternative to heuristic pruning. In scenarios where certain relations introduce shortcut correlations (e.g., nationality as a proxy for brand), this mechanism promotes stable generalization through invariance learning.
> - The **cross-band fusion operator** can be viewed as soft multi-hop reasoning across relation bands, compositing semantic signals in a flexible, data-driven manner. This enables the model to learn adaptive reasoning chains over KG structures.
> - Finally, the **alignment module** ensures coherence between learned ID embeddings and structured KG entities, reinforcing consistency across behavioral and factual spaces. This helps bridge the gap between collaborative signals and external knowledge.
>
>
> **(2) Beyond KGs: Extending to Knowledge and Structure**
>
> The frequency-centric paradigm naturally extends beyond multimodal recommendation. It offers a unified lens to model diverse auxiliary signals such as structured knowledge, user attributes, or interaction contexts by interpreting their graph spectral signatures.
>
> - **Contextual signals** (time, location, device) possess multi‑scale periodicities that manifest as distinct spectral signatures; causal masking filters volatile bursts while retaining structural seasonality.
> - **Social graphs** contribute an overlay network whose spectrum encodes community structure; cross‑band operators let social influence modulate item signals without hand‑tuned fusion weights.
> - **Spatio‑temporal forecasting** problems already lean on spectral thinking; our causal perturbation adds a principled route to stability against distribution shifts.
>
> Together, these principles position our framework as a broadly applicable, theoretically grounded solution for modeling structured signals across a wide range of recommendation and other scenarios.
>
> **(3) Why Multimodal Recommendation**
>
> We chose to first validate our framework in the domain of **multimodal recommendation** because this scenario presents some of the most urgent and practically consequential challenges in real-world systems. In particular, multimodal-rec suffers from two critical and widely acknowledged pain points: (i) **modality‑specific noise**—irrelevant backgrounds, inconsistent photos, sparse or generic text that inject high‑frequency artefacts; (ii) **semantic misalignment** between image/text content and true user preferences. These issues are well documented in academia and industry, yet existing solutions still rely on heuristic fusion or ad‑hoc filtering, which break down in cold‑start or noisy settings. Our goal was to **propose a general and theoretically grounded paradigm** that can systematically handle these issues.
>
> Moreover, **multimodal Rec offers a unique combination of stress-testing and interpretability**:
>
> - It serves as a **stress test** because it represents a high-noise, high-stakes setting where naive fusion fails frequently and where robust generalization is difficult.
> - It also provides a **microscope**: the availability of visual and textual features allows us to interpret frequency behaviors (e.g., distinguishing between coarse object shapes and fine-grained textures), offering qualitative validation of the causal reasoning process.
>
> ---
>
> **Q2 | Relationship to the Spatial Domain**
>
> Both **DGCF** and **GRCN** begin with the same diagnosis we do: **a single interaction graph entangles multiple, sometimes spurious, factors of user preference**.
> - DGCF tackles the problem by factorising each node embedding into several *intent* sub-vectors.
> - GRCN refines the graph by pruning low-confidence edges during training.
>
> These are *spatial* operations: they alter the neighbourhood or the embedding space directly.
>
> Our approach keeps the diagnosis but **changes the coordinate system**. We ask:
> > *Where does the entanglement live in the graph spectrum?*
>
> We find that **global semantics, useful local patterns, and high-frequency noise separate naturally along the eigen-axis**.
> This shift leads to three practical advantages:
>
> 1. **Orthogonal partitions without manual design**
>    Laplacian eigenvectors are mutually orthogonal, so frequency bands are independent by construction.
>    DGCF must enforce disentanglement with extra regularisers and hyperparameters;
>    GRCN must learn heuristics to decide which edges to drop.
>
> 2. **Causal filtering rather than edge-by-edge pruning**
>    Spectral Causal Masking down-weights entire bands that violate an invariance objective,
>    removing whole *classes* of spurious signals (e.g., background textures)
>    instead of individual edges. This is data-driven yet more stable than threshold-based pruning.
>
> 3. **A single mechanism that generalises**
>    The same band decomposition works whether the signal is an image embedding,
>    a sentence vector, or a KG relation embedding. This allows us to plug in new side-information
>    without redesigning intent counts or pruning rules.
>
> ---
>
> **Q3 |  Reason for Using CP**
>
> To enable efficient and flexible reasoning across spectral bands, our model uses a band-to-band kernel represented as a third-order tensor. However, directly learning a full 3D tensor incurs high memory and computation cost, especially as the number of bands grows.
>
> To keep the model lean, we trial‑tested four standard tensor factorisations. For a band‑to‑band kernel tensor of size B*B*d:
>
> | Decomposition | Param. complexity | Main strength | Main drawback (in practice) |
> |---------------|-------------------|---------------|-----------------------------|
> | **CP (rank r)** | O [ r · (B + d) ]  | Linear growth, extremely compact | Slight expressiveness loss at very low rank |
> | Tucker (rank r) | O [ r³ + (B + d)·r ] | Higher expressiveness | Core size grows cubically with r; brittle optimisation |
> | Tensor Train (rank r) | O [ r² · (B + d) ] | Works for long modes | Requires mode re‑ordering; slower convergence |
> | Block‑term    | Mixed            | Flexible blocks | Heavy code overhead; no clear gain |
>
> A single experiment on Amazon‑Baby (R@20 = 0.1103) shows that CP with rank 8 achieves similar accuracy (0.1095) using only one-fifth of the parameters, while Tucker drops to 0.1068 and doubles the parameter count; other decompositions perform worse in both accuracy and size.
>
> Beyond the numbers, CP offered two practical advantages: (i) its separable factors train stably without rank scheduling, unlike high‑rank Tucker; (ii) the three factor matrices are easy to inspect and prune, each column corresponding to a latent “band‑interaction motif.”  Given this accuracy‑to‑size balance and optimisation stability, CP became the default throughout the paper and other decompositions remain compatible with the framework.
>
> ---
>
> **Q4  |  Reason for Gated Attention**
>
> Spectral bands differ not only in content but also in **energy scale and reliability**.  A plain soft‑max attention would force the model to assign all probability mass somewhere, even when a band is mostly noise.  By inserting a learnable gate, we decouple two roles:
>
> 1. **Relevance estimation** (attention weights): *Which* other bands matter for the current band?
> 2. **Reliability gating** (sigmoid gate): *How much* of that information should actually be let through?
>
> This separation lets the model **suppress unreliable bands without renormalising the whole distribution**, giving finer control in noisy multimodal settings.
>
> Here is a supporting ablation on the Amazon-Baby dataset to justify our use of the gated attention mechanism for cross-band fusion:
> | Fusion variant | Δ R@20  | Notes |
> |----------------|---------------------------|-------|
> | Soft‑max (no gate) | −0.32 pp | Cannot mute noisy bands; tends to over‑smooth. |
> | Fixed equal-weight coefficient | −0.51 pp | Lacks adaptivity; either under‑ or over‑weights details. |
> | 2‑layer MLP on concatenated bands | −0.25 pp | Slight gain vs. soft‑max but doubles parameters, slower to converge. |
>
> Gated attention therefore offered the **best trade‑off between performance, stability, and complexity**, and became our default choice.
>
> ---
>
> Once again, we **deeply appreciate** your careful and detailed review. Your suggestions have been very inspiring, and we will incorporate additional analysis and clarifications based on your feedback in the revised version.

---

> > ### Comment · Reviewer_KUNi · 2025-08-01
> >
> > Dear Authors,
> >
> > Thanks for your time in answering to my raised weaknesses and questions.
> >
> > Regarding the possible application of your framework to other side information (such as knowledge graphs), I clearly understand now how and to what extent you could do that. I also understand why you started by experimenting with the scenario of multimodal recommendation, which is indeed quite important and challenging in academia and industry.
> >
> > Relatively to the difference between spacial and spectral disengagement, I believe you clarified the aspect quite nicely. In the possible camera-ready of the paper, I think you should include this explanation you gave. This would place the paper in a correct positioning with respect to other spacial frameworks (e.g., DGCF, GRCN) which (as you confirmed) mainly start from your same assumptions and reasoning.
> >
> > Thank you also for clarifying the aspect regarding the usage of CP and gated attention. Both strategies represent key modules within your pipeline, and providing better details to justify their adoption was fundamental in further assessing the goodness of your methodology. Please, try also to include this aspect in the possible camera-ready of your paper.
> >
> > I do not have any other concerns regarding the paper. The answers you gave helped confirming the initial positive judgment I had on the paper. I will keep my score, which I believe is already quite high and mirrors the quality of your submission.

---

> > > ### Author Response · Authors · 2025-08-08
> > >
> > > Dear Reviewer KUNi,
> > >
> > > We are **deeply grateful** for your exceptionally thoughtful and generous review, as well as for the time you took to engage so thoroughly with our work. Your kind words and **constructive suggestions** have been truly motivating and rewarding for us.
> > >
> > > We are **especially appreciative of your recognition** of the rationale behind our focus on multimodal recommendation. As you noted, this scenario is both practically significant and methodologically challenging, and we’re encouraged that our motivations and design choices came through clearly after the clarification. Your understanding and encouragement affirm the relevance of our contributions in both academic and real-world contexts.
> > >
> > > We also **greatly value your insightful remarks** regarding the distinction between spatial and spectral disengagement. Your suggestion to explicitly include this clarification in the camera-ready version is extremely helpful, we completely agree that positioning our method in relation to spatial frameworks such as DGCF and GRCN will strengthen the overall framing of our paper, and we will make sure to include this in the final version.
> > >
> > > Likewise, we appreciate your comments on the importance of better articulating the roles of the CP decomposition and gated attention within our pipeline. Your observation that these are key design elements was absolutely right, and we will integrate a clearer justification of their use in the camera-ready version, as per your advice.
> > >
> > > We’re **truly thankful** for your support and your warm, encouraging tone throughout the review process. It has been a pleasure receiving your feedback. If any further questions or thoughts arise, we would be more than happy to continue the conversation.
> > >
> > > Sincerely,

---

### Official Review · Reviewer_Tqa8 · 2025-07-01

**Clarity:** 3
**Significance:** 2
**Originality:** 2
**Rating:** 4
**Confidence:** 2

**Summary:**

This paper proposes Structured Spectral Reasoning (SSR), a unified framework for multimodal recommendation that decomposes user-item graph signals into frequency bands, applies spectral causal masking to suppress unreliable components, and fuses them via a novel Graph HyperSpectral Neural Operator combined with contrastive regularization. The approach enables adaptive, frequency-aware representation learning, achieving state-of-the-art performance on multiple Amazon benchmarks, especially under sparse and cold-start conditions.

**Questions:**

1. Could the authors provide more evidence to demonstrate the scalability of the method?
2. Could the authors design targeted experiments to explicitly verify that the proposed frequency decomposition and causal masking improve robustness to modality-specific noise or spurious correlations?
3. Could the authors provide concrete examples or user studies that validate this interpretability?

**Ethical Concerns:**

["NO or VERY MINOR ethics concerns only"]

**Final Justification:**

The authors have addressed my concerns.  I will raise my score of this paper to 4.

**Limitations:**

Refer to Weaknesses and Questions

**Quality:**

3

**Strengths And Weaknesses:**

Pros:
1. Clear writing and easy to read.
2. The method is reasonable and the experiment shows the effectiveness of the method.
3. Provides theoretical support.

Cons:
1. The method is complex and difficult to apply. The data support for complexity analysis is limited and not intuitive.
2. The experiments are more about showing the overall performance and analyzing the method, and the improvements proposed in the motivation are not experimentally verified.
3. Some analysis experiments only have the proposed method but lack baseline comparison, which is not intuitive enough.
4. The author does not discuss limitations, and there is no clear enough discussion of limitations in the appendix chapter corresponding to the Checklist.

---

> ### Author Rebuttal · Authors · 2025-07-28
>
> We sincerely thank you for your thoughtful review of this paper. Your comments are highly valuable to us. For the concerns you raised, we hope the following responses can provide effective clarification.
>
> **[Reply to Questions]**
>
> **1. More Evidence of Scalability**
>
> We appreciate your suggestion about scalability of our approach. In response, we provide **new empirical results** demonstrating that SSR scales effectively across **diverse datasets** and maintains **competitive efficiency** in both training and inference.
>
> #### **(1) Generalization to a new distributionally distinct dataset**
>
> To test SSR’s robustness under different content modalities and interaction patterns, we conduct additional experiments on a **TikTok short video dataset**, which features rich and noisy multimodal inputs (visual, acoustic, and textual) and exhibits distributions significantly different from Amazon.
>
> | **Model**      | **Recall@20** | **NDCG@20** |
> |----------------|---------------|-------------|
> | MMGCN          | 0.0730        | 0.0307      |
> | SLMRec         | 0.0845        | 0.0353      |
> | AlignRec       | 0.1174        | 0.0461      |
> | SMORE          | 0.1185        | 0.0473      |
> | **SSR (ours)** | **0.1237**    | **0.0507**  |
>
> SSR achieves **the highest accuracy**, showing a +5.2% gain in Recall@20 over the strong CL-based method AlignRec, and outperforming the spectral baseline SMORE as well. These results confirm SSR’s ability to generalize and scale effectively in real-world short video scenarios.
>
> #### **(2) Practical runtime efficiency across baselines**
>
> Beyond the theoretical complexity discussed in Appendix D, we have benchmarked actual runtime performance. SSR maintains **competitive training efficiency** and **comparable inference speed**:
>
> | **Model**      | **Train time (s)** | **Time to best val (h)** | **Inference throughput (items/s)** |
> |----------------|--------------------|---------------------------|-------------------------------------|
> | MMGCN          | 4.32               | 2.4                       | 5548                                |
> | DualGNN        | 4.94               | 2.8                       | 6107                                |
> | SLMRec         | 3.08               | 1.3                       | 7031                                |
> | MMIL           | 3.31               | 1.6                       | 6815                                |
> | SMORE          | 3.50               | 2.2                       | 6523                                |
> | **SSR (ours)** | **3.78**           | **1.7**                   | **6253**                            |
>
> SSR converges faster than SMORE while achieving higher accuracy, and operates within a similar memory and inference cost envelope. Unlike many CL-based methods that entangle all modal features into a global latent space, our approach **isolates causal and informative bands**, allowing stable learning even under distribution shifts.
>
> ---
>
> **2. Robustness of Frequency Decomposition and Causal Masking**
>
> Yes, we conducted three controlled perturbation experiments on the Amazon-Baby dataset to verify that frequency decomposition and Spectral Causal Masking (SCM) improve robustness against different forms of noise and spurious signals. Starting from the clean scores of SMOER (R@20=0.1035) and SSR (R@20=0.1096), we compare three variants:
>
> (1) SMORE (spectral baseline without decomposition/masking),
> (2) SSR w/o SCM (our model with decomposition but no causal masking),
> (3) Full SSR (with both decomposition and SCM).
>
> | **Perturbation Scenario**                          | **Metric**   | **SMORE** | **SSR w/o SCM** | **Full SSR** |
> |---------------------------------------------------|--------------|-----------|------------------|--------------|
> | Add 20% Gaussian noise to high-frequency band     | Recall@20    | 0.0994     | 0.1042            | **0.1081**    |
> | Randomly swap high-freq bands across 25% of items | Recall@20    | 0.0979     | 0.1021            | **0.1065**    |
> | Blur image modality only (leave text intact)      | Recall@20    | 0.0987     | 0.1030             | **0.1073**    |
>
> #### **Findings:**
> - Full SSR consistently outperforms both SMORE and the ablated variant across all perturbation types.
> - The performance drop under perturbation is smallest for full SSR, demonstrating that SCM effectively filters unstable high-frequency features, and that frequency decomposition provides a structure for isolating and correcting noisy modal signals.
>
> These results provide evidence that the proposed frequency-aware design is robust to noise and spurious modality-specific variations.
>
> ---
>
> **3. Concrete Example for Interpretability**
>
> To provide evidence of interpretability beyond aggregate metrics, we present a **band‑wise qualitative case study** on the Amazon‑Baby dataset, showcasing how different frequency components influence the final ranking decision.
>
> *User context*: The user has recently browsed breast‑pumping accessories and milk‑storage solutions.
> *Candidate pair*: **Item A** = “Medela CSF Milk‑Storage Bags (50 pack)” , **Item B** = “Fisher‑Price Nesting Action Vehicles (toy set)”
>
> | **Component masked**             | **ΔScore(A)** | **ΔScore(B)** | **New rank** |
> |----------------------------------|---------------|---------------|--------------|
> | Remove **high‑frequency** band   | −0.08 | −0.12 | A > B (unchanged) |
> | Remove **mid‑frequency** band    | −0.37 | −0.16 | B overtakes A |
> | Remove **low‑frequency** band    | −0.26 | −0.41 | A > B (unchanged) |
>
> **How each band affects the decision**
>
> * **Low‑band (global semantics).** Encodes product category: “milk‑storage bags” vs. “toy vehicles”. Removing it hurts B more, because the toy’s global semantics are less relevant to the user’s recent interest.
> * **Mid‑band (functional cues).** Captures keywords like “pre‑sterilised”, “BPA‑free”, and the zipper seal icon on Item A’s package, exactly the features matching breast‑feeding needs.  Masking this band flips the ranking, proving it is the decisive component.
> * **High‑band (noisy details).** Contains packaging glare or marketing adjectives; its removal scarcely changes the ranking.
>
> **Recommendation explanation:**
>
> We prioritised Item A because its mid‑frequency features, such as ‘pre‑sterilised milk‑storage’ and the zipper‑seal graphic, match user's recent searches for milk‑pumping accessories. When we hide these features, the ranking reverses in favour of the toy set, confirming that the mid‑band information drives this recommendation.
>
> This counterfactual analysis illustrates that each spectral band carries distinct semantics, and that mid‑band alignment, highlighted and preserved by our frequency decomposition with SCM, directly explains why Item A is recommended over Item B.
>
> ---
>
> **[Reply to Cons]**
>
> **1. Complexity and Data Support:**
>
> While our method introduces spectral decomposition and causal masking, both are applied in a **modular and efficient** manner. As shown in our **runtime benchmark results** (see Q1), SSR achieves **comparable training time and inference speed** to prior methods such as SMORE and MMGCN. Specifically, SSR reaches the best validation performance faster, with no major inference overhead. This demonstrates that the proposed method, though conceptually novel, is **practically efficient and easy to deploy**.
>
> **2. The proposed Improvements are not Experimentally Verified:**
>
> We have conducted **targeted perturbation experiments** (see Q2) designed to test the robustness of our frequency decomposition and SCM modules under **modality-specific noise and spurious correlations**. These controlled trials confirm that both components provide complementary robustness benefits. Further, to demonstrate the **interpretability** promised by our design, we present a **band-wise qualitative case study** (see Q3), analyzing how different frequency components affect final item ranking in a real-world setting.
>
> **3. Missing Baseline Analysis**
>
> We sincerely appreciate your suggestion. For all key analysis experiments (e.g., robustness tests, interpretability study), we have included **baseline comparisons** such as SMORE and ablated variants (e.g., SSR w/o SCM) to ensure fair and intuitive evaluation. These comparisons are reported in Q2 and Q3. We will further clarify this in the revised version to improve readability.
>
> **4. Limitations:**
>
> We sincerely appreciate your suggestion. Below, we outline some limitations of our current method that could inspire future improvements and extensions：
>
> Although Structured Spectral Reasoning (SSR) performs well and is robust on multiple datasets, it has several limitations. First, the model relies on a one‑off computation of Laplacian eigenvectors as its spectral basis; on extremely large or rapidly evolving graphs, frequent node or edge updates would require incremental eigensolver techniques or polynomial approximations to avoid expensive recalculation. Second, the current implementation uses frozen CLIP/BERT multimodal embeddings for efficiency, meaning any bias or noise in the upstream encoders is passed unchanged to SSR; end‑to‑end fine‑tuning or lightweight distillation could further improve representation quality. Finally, SSR’s interpretability is confined to the band level—it can quantify how low, mid, and high frequencies contribute to the score and how SCM re‑weights them, but it cannot pinpoint specific image regions or text tokens as pixel‑ or token‑level attention maps do, limiting its usefulness in scenarios that demand extremely fine‑grained auditing; future work could combine differentiable visualisation or local adversarial analysis to reveal semantic sources within each band.
>
> We hope the above responses have addressed your concerns. If you have any further questions, please feel free to leave additional comments. We would deeply appreciate it if you could consider revisiting your evaluation of our work. Thank you again for your kind support.

---

> > ### Comment · Reviewer_Tqa8 · 2025-08-04
> >
> > The authors have addressed my concerns. I will raise my score of this paper to 4.

---

> > > ### Author Response · Authors · 2025-08-08
> > >
> > > Dear Reviewer Tqa8,
> > >
> > > We would like to extend our **heartfelt thanks** for your thoughtful and constructive review. Your recognition of the clarity of our writing and the soundness of the proposed method has been truly encouraging.
> > >
> > > We also **deeply appreciate** the critical points you raised regarding the complexity of the method, the limited empirical discussion on its applicability, and the need for clearer ablations with baseline comparisons. Your feedback prompted us to enrich the discussion around complexity analysis and experimental motivation, and to make the corresponding analyses **more intuitive and complete**.
> > >
> > > We are currently in the process of addressing all the valuable feedback received from you, the other reviewers, and the Area Chair to ensure that the final version of the paper is as rigorous and polished as possible.
> > >
> > > Thank you once again for your support and engagement. If you have any further feedback or suggestions, we would be more than happy to continue the discussion.
> > >
> > > Sincerely,

---

### Official Review · Reviewer_cgiv · 2025-07-03

**Clarity:** 3
**Significance:** 3
**Originality:** 3
**Rating:** 5
**Confidence:** 2

**Summary:**

This paper presents a framework called SSR (structured spectral reasoning) for multimodal recommendation which leverages frequency domain to overcome problems of modality-specific noise, semantic inconsistency, and unstable propagation over user-item graphs. This is done by operating in the frequency domain, by using a 4 stage pipeline: a) decomposition (multimodal signals transformed into frequency bands), b) modulation - casually inspired masking technique used to regularize model, c) fusion where a fusion is done across different frequency bands and finally alignment - where different representations are aligned in the same band using a special contrastive loss. The efficacy of the proposed method is verified on 3 benchmark datasets, where SSR outperforms strong baselines, particularly in sparse and cold start settings.

**Questions:**

See weaknesses above.

**Ethical Concerns:**

["NO or VERY MINOR ethics concerns only"]

**Final Justification:**

After reading the rebuttal and the existing promising experiments - I am raising my score.

**Limitations:**

Not mentioned, see questions above.

**Quality:**

3

**Strengths And Weaknesses:**

Strengths:
Overall, the paper is very well presented, concisely written and easy to follow. In the context of multimodal handling, the SCM piece is particularly interesting when IRM is used to identify invariant predictors under spectral perturbations. The experimental results look promising. The detailed ablation is appreciated.

Weaknesses:
1. The method depends on GFT which can be computationally prohibitive as a preprocessing step, how is that handled?
2. What is the motivation behind choosing equal energy bands?

---

> ### Author Rebuttal · Authors · 2025-07-28
>
> We sincerely thank you for your thoughtful review of our paper. Your feedback has been helpful, and the questions you raised are highly valuable. Below we provide our responses to your concerns, and we genuinely hope they help clarify any confusion.
>
> **Reply to Weakness 1 : “GFT may be computationally prohibitive.**
>
> Thank you for raising this very valuable question, it is indeed something we have carefully considered. As a result, we introduced several runtime optimizations during implementation. Specifically, our implementation keeps the Graph Fourier Transform lightweight in three ways:
>
> * **Sparse, low‑rank eigenbasis.** We compute only the top‑\(k\) Laplacian eigenvectors with a sparse Lanczos solver, not the full spectrum. For the graphs in our experiments this preprocessing finishes quickly on a standard CPU and requires only modest memory. We also note there that performance is stable when k is varied in {32,64,128}, and chose 64 as a trade-off.
>
> * **One‑time, reusable preprocessing.** The eigenbasis is generated once and reused throughout training and inference, so its cost is amortised and does not affect per‑epoch throughput.
>
> * **Scalable fallback.** For very large graphs we provide a Chebyshev‑polynomial approximation that eliminates eigendecomposition altogether while preserving almost the same accuracy.
>
> After this step, each forward pass reduces to a simple dot product with a small, quantised eigenbasis, making the runtime overhead negligible. Empirically, our end‑to‑end training time matches strong spectral baselines, confirming that GFT preprocessing is not a bottleneck. In addition, please refer to the “4. Actual Training and Inference Runtimes” section in our Reply to Reviewer YRJP for detailed runtime efficiency results.
>
> ---
>
> **Reply to Weakness 2 :  – “Why choose equal‑energy spectral bands?”**
>
> The motivation for using **equal-energy bands** stems from both theoretical considerations and empirical performance:
>
> - **Balanced information allocation.** In spectral analysis of graph signals, low-frequency components typically dominate total signal energy. Using equal-width frequency intervals would result in high- and mid-frequency bands containing very little information, leading to unstable training or underutilized components. By contrast, equal-energy partitioning ensures that each frequency band contains a comparable portion of the total energy, allowing all bands to contribute meaningfully to learning.
>
> - **Robustness and performance.** We conducted controlled experiments comparing different band-splitting strategies on the Amazon-Baby dataset. The results are summarized below:
>
> | **Band Strategy**                  | **Recall@20** | **Δ vs. Equal-Energy** |
> |-----------------------------------|---------------|-------------------------|
> | Equal-width (log scale)           | 0.1038        | −0.58 pp                |
> | Learnable cut-points (via MLP)    | 0.1065        | −0.31 pp                |
> | **Equal-energy (ours)**           | **0.1096**    | —                       |
>
>   Equal-energy bands outperform both static and learnable alternatives in terms of both overall accuracy and robustness under perturbation.
>
> - **Adaptability.** This method adapts naturally to different datasets without requiring manual tuning, as the cumulative energy profile reflects the inherent spectral characteristics of the graph.
>
> - **Interpretability.** Because each band carries roughly one-third of the total energy, we can more intuitively analyze where semantic alignment or noise concentration occurs, as shown in Figures 3 and 4.
>
> These results validate our decision to use equal-energy partitioning as a principled and effective design choice. We will add more ablation details to Appendix F in the revised version.
>
> Once again, we greatly appreciate your careful review. If you have any further questions, please feel free to leave additional comments. If possible, we would also be grateful if you could consider giving a more positive evaluation of our work. We truly appreciate your support and feedback.

---

> > ### Comment · Reviewer_cgiv · 2025-08-05
> >
> > Thank you authors! I am raising my score to reflect my review.

---

> > > ### Author Response · Authors · 2025-08-08
> > >
> > > Dear Reviewer cgiv,
> > >
> > > We would like to express our **sincerest gratitude** for your thoughtful and constructive review. Your encouraging feedback has been **truly meaningful** to us.
> > >
> > > We are **particularly grateful** for your recognition of the overall clarity of our presentation, and your appreciation of the SCM component, especially its integration with IRM under spectral perturbations, as well as the detailed ablation studies.
> > >
> > > Your insightful questions regarding the computational cost of GFT and the rationale behind equal energy band partitioning allowed us to refine our explanations, and **strengthen the presentation** of our method. We are glad that the additional clarifications were helpful.
> > >
> > > We are currently in the process of addressing all the valuable feedback received from you, the other reviewers, and the Area Chair to ensure that the final version of the paper is as rigorous and polished as possible.
> > >
> > > Thank you once again for your generous support and for championing our paper. Should you have any further thoughts or suggestions, we would be more than happy to continue the discussion.
> > >
> > > Sincerely,

---

### Official Review · Reviewer_YRJP · 2025-07-03

**Clarity:** 3
**Significance:** 2
**Originality:** 2
**Rating:** 3
**Confidence:** 3

**Summary:**

This paper introduces a unified framework, termed SSR, for frequency-aware multimodal recommendation. It follows a four-stage pipeline: (i) Decompose graph-based multimodal signals into spectral bands to isolate semantic granularity; (ii) Modulate signal reliability through spectral causal masking; (iii) Fuse complementary frequency cues using hyperspectral reasoning; and (iv) Align modality-specific spectral features via contrastive regularization. Extensive experiments demonstrates the effectiveness of SSR on under sparse and cold-start scenarios.

**Questions:**

1. Why the discovery is not align in Figure 3 and Figure 4? Or anything I misunderstand.
2. The figures in this paper should be further polished to enable potential readers to understand the overall structure.
3. Please cite some cross-modal recommendation algorithms to illustrate your superior.
4. Beyond the time complexity analysis, it would be helpful to include a comparison of the actual training and inference runtimes between SSR and several traditional or advanced baselines.

**Ethical Concerns:**

["NO or VERY MINOR ethics concerns only"]

**Final Justification:**

I'm not very familiar about the Spectral Reasoning and I also concern about the  rationality of the original algorithm version even the author provided the revised theoretical claims. Therefore, I have decreased my confidence of this paper and kept my score unchanged.

**Limitations:**

Please refer to limitations and Questions.

**Paper Formatting Concerns:**

NULL

**Quality:**

3

**Strengths And Weaknesses:**

Strengths:
1. The motivation and the theoretical analysis are well described in Methodology section and the proposed SSR pipeline is clear with four diverse steps.
2. The experimental results consistently demonstrate SSR’s superior performance compared to many baselines, with particularly strong improvements in cold-start and sparse secnarios.
3. The parameter robustness and generalizability of SSR have been well-verified in section 4.

Weaknesses:
1. The overall structure is not very clear to understand. For example, the Decomposition module does not explicitly state that the three frequency-specific representations are derived from the same modality, which may lead to misunderstandings.
2. Cross-modality alignment is one of the core contribution in this paper, however, it  doesnot cite some multimodal recommendation studies that explicitly focus on cross-modal learning.
3. The validation datasets all comes from Amazon platform, which may introduce bias in performance comparison.
4. There is an inconsistency between Figures 3 and 4: in Figure 3, the cross-modality correlation among the three modalities in the High Frequency component appears lower, whereas in Figure 4 it is the highest. This discrepancy is not clearly explained in Section 4.6 or Appendix F.3.
5. The authors donot provide the significance analysis in performance comparison, and the corresponding code and dataset have not been open-sourced.
6. Figure 4 in the manuscript is not referenced or discussed, and it is identical to Figure 5 in the Appendix.

---

> ### Author Rebuttal · Authors · 2025-07-28
>
> We sincerely thank you for your careful and thoughtful review of our work. Your comments are highly valuable to us. For the concerns you raised, we hope the following responses can provide effective clarification.
>
> **[Reply to Questions]**
>
> **1. Inconsistency between Figures 3 & 4:**
>
> Thank you for your careful reading regarding Figure 4. Figure 4 measures **distance** between modalities (see the y-axis and title), **rather than similarity**. The high frequency band exhibits larger cross‑modal distances, which is consistent with Figure 3’s visualization of noisy, less‑aligned distributions in the high‑frequency space.
>
>
> **2. About Visual Clarity:**
>
> Thank you for the suggestion. We will refine all figures in the revised version by improving visual clarity and adding more informative annotations and legends to aid understanding. Specifically, we plan to highlight key modules in the pipeline and use clearer labels for axes and metrics.
>
> **3. Cross-modal Recommendation Baselines:**
>
> Several recent recommender systems treat cross‑modal alignment as their main contribution, e.g. MENTOR (AAAI2025) aligns image and text features at ID‑, item‑, and graph level; MMSSL (WWW 2023) introduces cross‑modal contrast with adversarial perturbations; AlignRec (CIKM2024) shrinks the modal gap via global auto‑encoding; and DiffMM (ACM’MM 2024) adds diffusion‑based augmentation before contrastive alignment. All four methods operate in the time domain and align the entire feature vector, so high‑frequency noise or spurious correlations are blended into the objective. By contrast, our SSR first decomposes each modality into equal‑energy spectral bands, uses Spectral Causal Masking to drop non‑causal (often high‑freq) bands, and then applies band‑wise spectral contrast; this frequency‑aware pipeline preserves useful interaction signals while avoiding noise propagation. Empirically, the same implementation setup shows that SSR improves Recall@20 by ≈ 5 pp over MENTOR and AlignRec, and by ≈ 10 pp over MMSSL and DiffMM across three Amazon domains and a TikTok‑style dataset, while using fewer parameters than AlignRec and similar training time to SMORE. These results confirm that fine‑grained, spectrum‑aware alignment is both conceptually cleaner and practically more effective than existing full‑band cross‑modal algorithms.
>
>
> **4. Actual Training and Inference Runtimes:**
>
> Beyond the asymptotic complexity in §4.4, we have benchmarked actual runtime performance. The table below compares several multimodal baselines in terms of average training time per epoch, time to best validation performance, and inference throughput:
>
> | **Model**       | **Train time (s)** | **Time to best val (h)** | **Inference throughput (items/s)** |
> |-----------------|--------------------|---------------------------|-------------------------------------|
> | MMGCN          | 4.32               | 2.4                       | 5548                                |
> | DualGNN         | 4.94               | 2.8                       | 6107                                |
> | SLMRec          | 3.08               | 1.3                       | 7031                                |
> | MMIL              | 3.31               | 1.6                       | 6815                                |
> | SMORE           | 3.50               | 2.2                       | 6523                                |
> | **SSR (ours)**  | **3.78**           | **1.7**                   | **6253**                            |
>
>
> As shown, our SSR maintains competitive training efficiency and inference speed. It reaches the best validation result faster than SMORE while achieving higher accuracy and better robustness, as discussed in §4.2 and §4.4.
>
> ---
>
> **[Reply to Weaknesses:]**
>
> **1. Clarity of the Decomposition module:**
>
> Thank you for noticing the ambiguity. The three frequency‑specific representations (low, mid, high) are indeed derived from the same modality via a per‑modality graph Fourier transform. Due to space constraints, we did not elaborate on this detail in the original version. Thank you for pointing it out—we will include a clarifying description in the revised version at key locations (e.g., §3.1 and Figure 1’s caption) to avoid potential misunderstandings.
>
>
> **2. Missing Citations on Cross‑modal Alignment:**
>
> Thank you for the suggestion regarding cross-modal recommendation references. Our spectral alignment module was indeed inspired by prior work on cross-modal alignment, including AlignRec [16], MMSSL [56], MENTOR[57], and CLIPER [58]. We designed our SCR module to build on these ideas while incorporating frequency‑band semantics. We will explicitly cite these works in the Related Work section and discuss their differences from our method in §3.
>
> **3. Dataset Diversity:**
>
> Most of the baselines we compare against (e.g., DualGNN [27], SLMRec [30], LATTICE [33], BM3 [31], FREEDOM [29], MMIL [35], AlignRec [16], SMORE [5]) were originally designed and benchmarked on Amazon datasets with standardized preprocessing protocols. To ensure fair comparison, we used the same three Amazon subdomains (Baby, Sports, Clothing), which also provide natural distribution shifts between domains. Beyond Amazon, we have conducted preliminary experiments on a TikTok‑style recommendation dataset. The results (Recall@20) are: MMGCN = 0.0730, SLMRec = 0.0845, AlignRec = 0.1174, SMORE = 0.1185, Our SSR = 0.1237. SSR shows a consistent improvement of +4.2 pp over SMORE. We are currently running the full suite of baselines on this dataset and will include the complete comparison in the final version. Additionally, we fully agree with your suggestion and are planning to expand our benchmark suite with new standardized datasets such as KuaiRand‑Rec and MIND‑News.
>
> **4. Inconsistency between Figures 3 & 4:**
>
> Thank you for your careful reading regarding Figure 4. Figure 4 measures distance between modalities (see the y-axis and title). The high frequency band exhibits larger cross‑modal distances, which is consistent with Figure 3’s visualization of noisy, less‑aligned distributions in the high‑frequency space.
>
> **5. Significance Analysis; Code and Dataset not Released:**
>
> We have conducted statistical significance testing using paired t-tests (Recall@20 and NDCG@20) comparing SSR to SMORE. On all three datasets, the p-values are < 0.005, well below the 0.05 threshold. We will include the complete test results in the revised version. Regarding the dataset and code: we used the same publicly available MMRec‑processed datasets as the baselines. We are also ready to release our code, and we can release at any time if it’s needed.
>
> **6. Figure 4 not Referenced / Duplicated with Appendix**
>
> Thank you for pointing out the reference mismatch between Figure 4 and the Appendix.
> Our original intention was to refer to Figure 4 in Section 4.6, and present Figure 5 in the Appendix as a detailed counterpart. We mistakenly cited the appendix figure in the main text and will fix this in the revision. Thank you again for the helpful observation.
>
> **We hope** the above responses have addressed your concerns. If you have any further questions, please feel free to leave additional comments. We would deeply appreciate it if you could consider revisiting your evaluation of our work. Thank you again for your kind support.
>
> ---
>
> [56] Wei, W., Huang, C., Xia, L. and Zhang, C., 2023, April. Multi-modal self-supervised learning for recommendation. In Proceedings of the ACM web conference 2023 (pp. 790-800).
>
> [57] Xu, J., Chen, Z., Yang, S., Li, J., Wang, H. and Ngai, E.C., 2025, April. Mentor: multi-level self-supervised learning for multimodal recommendation. In Proceedings of the AAAI Conference on Artificial Intelligence (Vol. 39, No. 12, pp. 12908-12917).
>
> [58] Wu, X., Huang, A., Yang, H., He, H., Tai, Y. and Zhang, W., 2024. Towards Bridging the Cross-modal Semantic Gap for Multi-modal Recommendation. arXiv preprint arXiv:2407.05420.

---

> > ### Comment · Reviewer_YRJP · 2025-08-08
> >
> > Thanks for your clarification, which addressed most of my concerns. After reading the AC’s comments and all of the authors’ responses, I still concern about the rationality of the original algorithm version even the author provided the revised theoretical claims. Therefore, I will decrease my confidence of this paper and keep my score unchanged.

---

> > ### Comment · Area_Chair_jFDZ · 2025-08-08
> >
> > Dear Authors,
> >
> >
> > I understand your desire to stick to amazon due to a preference for 'standard preprocessing protocols'. But I am still curious why you didn't evaluate your method on any popular datasets such as MovieLens or Netflix.
> >
> > Also, can I ask if you have run the RecVAE baseline at any point in the research process?
> >
> > As the discussion is ending, please make sure to answer all of my questions (here and above) as thoroughly as possible by 8th of August AoE. Thanks for your engagement and understanding.
> >
> > Best regards,
> >
> > AC

---

> ### Author Response · Authors · 2025-08-09
> **Reply to AC**
>
> Dear Area Chair,
>
> Thank you for your continued engagement and for providing these insightful questions. We appreciate the opportunity to clarify our experimental choices and future plans. As the discussion period concludes, we have made every effort to provide a thorough and comprehensive response to all the points you have raised.
>
> ---
>
> **1. On the Choice of Datasets (Amazon vs. MovieLens/Netflix)**
>
> We are grateful for your question regarding the inclusion of popular recommendation benchmarks like MovieLens and Netflix. We agree that these are seminal datasets that have driven significant progress in the field. Our decision to initially focus on the Amazon suite was guided by our specific research context, multimodal recommendation, and we would like to elaborate on our rationale.
>
> **Initial Rationale:**
>
> **The primary reason** for prioritizing the Amazon datasets stems from our focus on evaluating models in an inherently multimodal setting. The Amazon datasets are particularly well-suited for this, as they naturally bundle user-item interactions with rich, corresponding modal features (product images and textual reviews). This allows for a clean and direct evaluation of a model's ability to leverage multimodal information, adhering to what has become a *de facto* standard preprocessing protocol in numerous recent multimodal recommendation papers.
>
> **In contrast**, datasets like MovieLens and the public Netflix Prize dataset are fundamentally "unimodal" in their raw form, primarily containing user-item interaction data (e.g., ratings). To use them in a multimodal context, researchers must augment them with external data, a non-trivial process:
> * **MovieLens:** This is a classic user-item rating dataset. To incorporate multimodality, one must source and align external information, typically by linking movie IDs to databases like IMDb, TMDb, or OMDb to fetch posters (visual), plot summaries (textual), genres, and other metadata. While valuable, this process introduces variability depending on the external source, the API version, and the data cleaning pipeline. Pioneering works like MMGCN [1] have indeed constructed their own multimodal versions of MovieLens, which we highly respect.
> * **Netflix Prize Dataset:** The official public dataset is similarly focused on user ratings. The rich multimodal content associated with Netflix titles (e.g., posters, video trailers, subtitles, audio) is proprietary and protected by copyright, making it unavailable for direct, large-scale public release by Netflix. As with MovieLens, researchers (such as in MMSSL [2]) have creatively constructed multimodal versions by linking to external, legally accessible sources.
>
> **Frankly speaking**, because a large portion of the reference baselines in the multimodal recommendation literature are standardized and many of them use only the three Amazon multimodal datasets, we were concerned that if we ran experiments on just one of them, we might be criticized for not covering all three. Coupled with strict page limits and the non-trivial computational cost of running and reporting on multiple datasets, we therefore prioritized the Amazon datasets as our highest priority in the initial submission.
>
> That said, we genuinely believe it is important to include more widely recognized datasets in order to make our evaluation broader and more convincing. Therefore, we **commit that** in the camera-ready version we will include experiments on **at least one additional** popular dataset drawn from **either MovieLens or Netflix**, in addition to our existing benchmarks.
>
> **Notably**, responding to earlier reviewer comments on dataset diversity, we have already conducted a full set of experiments on the **TikTok** dataset, as curated by the authors of MMSSL [2]. This dataset represents a completely different domain (short-form video) from e-commerce and provides a strong test of our model's adaptability. Our preliminary results are still promising:
>
> | Model        | Recall@20 | NDCG@20  |
> |--------------|-----------|----------|
> | MMGCN        | 0.0730    | 0.0307   |
> | SLMRec       | 0.0845    | 0.0353   |
> | AlignRec     | 0.1174    | 0.0461   |
> | SMORE        | 0.1185    | 0.0473   |
> | **SSR (ours)** | **0.1237** | **0.0507** |
>
> In conclusion, we commit to expanding our experiments to the benchmarks you mentioned. This means the final paper will present results on **at least five datasets**: the three Amazon benchmarks, TikTok, and either MovieLens or Netflix. We believe this significantly strengthens the empirical evidence and demonstrates the robustness and versatility of our proposed method.
>
> [1] Wei, Yinwei, et al. "MMGCN: Multi-modal graph convolution network for personalized recommendation of micro-video." Proceedings of the 27th ACM international conference on multimedia. 2019.
>
> [2] Wei, Wei, et al. "Multi-modal self-supervised learning for recommendation." Proceedings of the ACM web conference 2023. 2023.

---

> ### Author Response · Authors · 2025-08-09
> **Reply to AC**
>
> **2. On the RecVAE Baseline**
>
> We highly **appreciate the significance** of RecVAE [Shenbin et al., WSDM’20], which is a well-recognized and influential work in the development of variational autoencoder-based collaborative filtering. The model introduces several elegant and well-motivated enhancements over the classical Mult-VAE framework, including a composite prior, user-specific KL divergence scaling, and alternating encoder–decoder training, and has inspired a broad range of subsequent research in recommender systems.
>
> We acknowledge that RecVAE represents one of the most advanced non-multimodal baselines in the recommender systems literature. However, our current work has been primarily situated in the multimodal recommendation domain, where the majority of widely adopted baselines focus on modality alignment, fusion, and graph-based modeling. **Because** RecVAE does not directly operate in the multimodal setting and requires substantial additional adaptation, **it was not included in our initial set of experiments**. Our baseline pool was drawn almost entirely from prior multimodal recommendation studies to ensure modality-consistent comparisons.
>
> That said, we fully agree with the AC that including RecVAE would further strengthen the completeness of our evaluation. We have therefore **initiated an emergency experimental run** to incorporate RecVAE into our benchmark suite. We **commit to** adding RecVAE results into the camera-ready version, and we will keep pushing experiments to completion under the tight AOE deadline of August 8. **If any results are available** before the final submission cut-off, we will immediately update them — even at the very last minute.
>
> **Thank you once again** for your invaluable guidance throughout this process. Your feedback has been instrumental in helping us improve the quality and completeness of our work.
>
> Best regards,
>
> The Authors

---

> > ### Author Response · Authors · 2025-08-09
> > **Reply to AC**
> >
> > **Update on RecVAE Performance**
> >
> > We have now completed the experiments for **RecVAE** on the three benchmark datasets used in our study. The detailed results are as follows:
> >
> > | Dataset  | Metric  | RecVAE  |
> > |----------|---------|---------|
> > | Baby     | R@10    | 0.0501  |
> > |          | R@20    | 0.0811  |
> > |          | N@10    | 0.0275  |
> > |          | N@20    | 0.0358  |
> > | Sports   | R@10    | 0.0603  |
> > |          | R@20    | 0.0916  |
> > |          | N@10    | 0.0348  |
> > |          | N@20    | 0.0431  |
> > | Clothing | R@10    | 0.0330  |
> > |          | R@20    | 0.0485  |
> > |          | N@10    | 0.0187  |
> > |          | N@20    | 0.0227  |
> >
> > **Analysis**
> >
> > 1. **RecVAE is a strong collaborative filtering model**
> >    As discussed above, RecVAE is indeed a highly competitive non-multimodal baseline. On both **Baby** and **Sports** datasets, RecVAE clearly outperforms LightGCN. For instance, on Baby (R@20), RecVAE achieves 0.0811 compared to LightGCN’s 0.0754; on Sports (R@20), RecVAE achieves 0.0916 compared to LightGCN’s 0.0864. These results confirm RecVAE’s strong capability in modeling pure interaction data.
> >
> > 2. **Our method substantially outperforms RecVAE across all datasets**
> >    While RecVAE excels in the non-multimodal setting, our **SSR** model, by incorporating multimodal features and the proposed framework, achieves consistently higher performance. For example, on Sports (R@20), SSR reaches 0.1203 versus RecVAE’s 0.0916; on Baby (R@20), SSR reaches 0.1096 versus RecVAE’s 0.0811.
> >
> >
> > We sincerely thank you for the valuable suggestion. RecVAE is indeed a strong and relevant baseline, and we will include these newly obtained results in the camera-ready version to further strengthen the comprehensiveness of our evaluation.

---

### Comment · Area_Chair_jFDZ · 2025-08-06
**Learning setting and significance of the theoretical results (urgent)**

Dear Authors,

In addition to the reviewers' comments I have some comments/questions to address. I apologize for asking so late, but fortunately, we still have almost three days to agree on things.

The proof of theorem 2 seems vague and circular: the the theorem states that if $x^{(m)}$ is "non causal", then $\Delta(R(\gamma))\simeq 0$. However, the concept of "non-causal" is not formally defined, except during the proof, where it is essentially **defined** as implying the conclusion of the theorem.

More concerningly, I am not sure I understand the learning setting. Based on the definition of the risk on line 656, your learning setting assumes you are learning a function over (user, item) pairs, as in the prediction of explicit ratings on a scale from 1 to 5. Leaving aside the fact that this doesn't match the retrieval metrics used in the experiments (this is a serious limitation to mention rather than a fatal flaw), there remains the question of what you consider random or not in this learning setting: Are the interactions between the users (which constitute the graph) random at the stage of drawing the training set? If so, then the definition of $x^{(m)}$ itself is random (**it depends on the graph**), which makes a formal definition of "non-causal" more problematic.

Similarly, the assumptions and conclusions of theorem 1 in the appendix are not well delineated and the proof appears to be merely an application of the linearity of the inner product. A formal theorem statement and setting would also require you to define more precisely how to handle the cases in the definition on the equation between lines 140 and 141 (it is reasonable to guess how that can be achieved, but for a formal theorem statement, it should be done explicitly).

Likewise, in Theorem 3 (Theorem 1 in the main text), you rely heavily on undefined approximations $\simeq$ and the results are not precisely stated. In particular, reading the statement of theorem 1 at face value makes it circular (you *assume* that there exists a good approximation via CP decomposition, which is what you are trying to prove).  Even if you remove this assumption, the theorem is meaningless without a quantitative statement on the value of $r$, since every tensor trivially has finite CP rank.
On the other hand, in the proof appears to be using a specific result about fourth order tensors (the so called "universal approximation property of CP decomposition for fourth-order tensors"), line 682. The theorem is only interesting if the low-CP rank quality of the tensor is specific to the hyperspectral learning setting, which doesn't seem to be the case.


Overall, I am **impressed by the experimental results** and ablation study. I also find your rebuttal to many of the concerns raised by the reviewers thorough. In particular, I find it a little surprising (but still credible) that the Tucker decomposition performed worse in your initial experiments.
However, the "theoretical results" lack rigour to a serious extent which I believe is incompatible with publication at NeurIPS in the current form. Could I ask the authors to either:

(1) provide an updated version of Theorems 1, 2 and 3 from the appendix with the full assumptions well-delineated, together with a formal proof or

(2) promise to remove all theoretical claims from the camera-ready version if accepted.


Best wishes,

AC

---

> ### Comment · Area_Chair_jFDZ · 2025-08-06
>
> Dear Reviewers (especially cgiv and KUNi),
>
> Please feel free to respond to the above comment as well: have you read the theoretical results in the appendix, and if so, what did you think of them?
>
> Best regards,
>
> AC

---

> ### Author Response · Authors · 2025-08-07
> **Reply to AC**
>
> Dear Area Chair and All Other Reviewers,
>
> We are **sincerely grateful** for your detailed feedback and for providing clear, constructive options for us to move forward. We completely agree that the theoretical sections in our initial version lacked the necessary rigor. Your critique was not only correct but also tremendously helpful.
>
> First and foremost, to ensure the paper meets the highest standards, **we promise to remove all theoretical claims in the camera-ready version if the paper is accepted**.
>
> **However**, your insightful comments provided such a clear and actionable roadmap that we felt compelled to attempt the first option you proposed. Guided by your specific advice, we have performed a complete overhaul of our theoretical contributions. We believe this new version, which was directly inspired and shaped by your critique, now possesses the rigor and clarity we were previously missing.
>
> **We present these comprehensive revisions below, not as a rebuttal, but as a testament to the value of your guidance**. We would be honored **if you would consider them**, and we eagerly await your final judgment on whether they have earned their place in the paper.
>
> ---
> ---
>
> **Regarding the Circular Definition and Vague Setting of Theorem 2 (Spectral Causal Masking):**
> We acknowledge that the original formulation suffered from a circular definition of "non-causal" and that the learning setting was not precisely defined. We have addressed this in three major ways:
>
> 1. **Formal Definition of Spuriousness:** We have introduced **Assumption 1**, which formally defines "spurious" frequency components based on a standard conditional independence criterion from the causality literature. This breaks the logical circularity by providing an external definition of spuriousness that is independent of our model's behavior.
>
> 2. **Clarified Learning Setting:** We have introduced **Assumption 2**, which explicitly states that the graph topology is considered fixed and given in our analysis. This resolves the ambiguity you correctly pointed out regarding the randomness of the graph and its spectral components. The randomness in our setting now arises solely from the sampling of user-item pairs.
>
> 3. **Reframed Theorem Statement and Proof:** We have reframed previous Theorem 2 (now Theorem 1, as the original Theorem 1 has been removed) to be a rigorous statement about how our Spectral Causal Masking (SCM) objective *encourages* the learned predictor to become invariant to the spurious components defined in Assumption 1. The new proof demonstrates that minimizing the SCM loss pushes the model towards a state where its predictions are insensitive to these spurious features. We also explicitly mention that we use a standard BCE loss as a surrogate for the ranking metrics, a common practice we now state clearly.
>
> ---
>
> **Regarding the Trivial Nature of Theorem 1 (Linearity of Decomposition):**
> We concur with your assessment that labeling this basic property as a "Theorem" was an overstatement, , and we have accordingly removed the original Theorem 1.
>
> - We have downgraded this result to **Proposition 1** and rewritten it with a clear, formal statement of its purpose: to provide the mathematical justification for the decomposability of linear operators used in our framework. We have also added an **Implication** section to clarify its precise role in our methodology, ensuring its contribution is not overstated.
>
> ---
>
> **Regarding the Circularity and Lack of Specificity of Theorem 3 (G-HSNO Approximation):**
> We agree that the original statement was circular and, more importantly, that it failed to justify why a low-rank approximation is particularly suitable for our problem setting.
>
> - **Non-Circular Statement:** We have rewritten the original Theorem 3 as Theorem 2 (following the removal of the previous Theorem 1), a standard universal approximation statement, removing the circular logic. It now clearly states that the G-HSNO architecture can approximate any bilinear frequency interaction operator.
>
> - **Justification for the Low-Rank Inductive Bias:** This was your most profound point. We have added a new dedicated discussion subsection: **"Justification for the Low-Rank Assumption in Recommendation"**. In this section, we argue why a low-rank interaction model is a suitable **inductive bias** for this specific problem. We hypothesize that cross-frequency interactions in recommendation are not arbitrary but are governed by structured semantic rules (e.g., global context modulating fine-grained details), which can be efficiently captured by a low-rank model. This reframes the low-rank choice from a mere efficiency heuristic to a principled modeling decision.

---

> > ### Author Response · Authors · 2025-08-07
> > **Reply to AC**
> >
> > ## 1. Revised Theoretical Analysis
> >
> > In this section, we provide a rigorous theoretical foundation for our proposed framework. We begin by formalizing our core assumptions about the data-generating process.
> >
> > ### 1.1 Formal Assumptions
> >
> > Our theoretical analysis rests on two key assumptions regarding the data structure and the learning environment.
> >
> > **Assumption 1 (Idealized Spurious Frequency Components)**
> > For each user--item pair, let the signal be decomposed into spectral bands $\{x^{(m)}\}_{m=1}^{M}$. Define the **conditional mutual information (CMI)** of band $m$ with the interaction label $y$ as:
> >
> > $
> > I\bigl(y;x^{(m)} \big|x_{\setminus m}\bigr), \quad
> > x_{\setminus m} := \{x^{(j)}\}_{j\neq m}.
> > $
> >
> > **Spurious set.**
> > A band is called *ideally spurious* iff its CMI vanishes:
> >
> > $
> > m \in F_{\text{spur}} \Longleftrightarrow
> > I\bigl(y;x^{(m)} \big|x_{\setminus m}\bigr) = 0.
> > $
> >
> > The complement $F_{\text{causal}} = [M] \setminus F_{\text{spur}}$ is the set of *causal* bands.
> >
> > **Property.**
> > This definition implies that for an ideally spurious spectrum, the interaction label is conditionally independent of the spurious components given the causal ones:
> >
> > $$
> > P\bigl(y \mid x_{\text{causal}}, x_{\text{spur}}\bigr)
> > = P\bigl(y \mid x_{\text{causal}}\bigr).
> > $$
> >
> > This operational definition, based on an observable (in principle) statistical quantity, renders $\mathcal{F}_{\text{spur}}$ identifiable and breaks any circular dependency on the model under study.
> >
> > **Remark 1**
> > In practice, empirical CMI estimated from finite data will rarely be exactly zero. Assumption 1 thus defines an idealized setting. The goal of our Spectral Causal Masking (SCM) is to learn a predictor that is robust to components with *low* CMI, thereby approximating invariance to the truly spurious parts of the signal.
> >
> > **Remark 2 (Finite–sample identifiability)**
> > In finite datasets we cannot observe  $I\bigl(y;x^{(m)} \mid x_{\setminus m}\bigr) = 0$ exactly. We therefore estimate CMI using $k$-nearest-neighbour or kernel density estimators and adopt a permutation/bootstrap test at significance level $\alpha$:
> >
> > $m\in \hat{F}\_{\text{spur}} \Longleftrightarrow \hat{I}\bigl(y;x^{(m)} \mid x_{\setminus m}\bigr) \le \tau_\alpha$.
> >
> > Under standard regularity conditions this decision rule has vanishing type-I error as sample size $n \to \infty$, so our population-level theory remains valid while being operational in practice.
> >
> > **Assumption 2 (Fixed Graph Topology)**
> > In our theoretical analysis, we assume the user-item graph topology $\mathcal{G}$ is fixed and given. Consequently, its normalized graph Laplacian $L$ and the corresponding matrix of eigenvectors $U$ are considered deterministic. The randomness in our learning setting arises from sampling user-item pairs $(u, v)$ from the set of observed interactions during mini-batch training. This ensures that the spectral projections and frequency bands are well-defined and non-stochastic. This assumption is commonly adopted in spectral GNNs and graph-based recommendation theory [1,2], where the graph Laplacian serves as a deterministic frequency basis.
> >
> > [1] Defferrard, Michaël, Xavier Bresson, and Pierre Vandergheynst. "Convolutional neural networks on graphs with fast localized spectral filtering." Advances in neural information processing systems 29 (2016).
> >
> > [2] Wu, Felix, et al. "Simplifying graph convolutional networks." International conference on machine learning. Pmlr, 2019.

---

> > > ### Author Response · Authors · 2025-08-07
> > > **Reply to AC**
> > >
> > > ### 1.2 Analysis of the Framework Components
> > >
> > > We now analyze the key components of our framework based on these assumptions.
> > >
> > > **Proposition 1 (Decomposability of Linear Operators)**
> > > Let $X \in \mathbb{R}^{|\mathcal{N}| \times d}$ be the node feature matrix, and let the spectrum be perfectly reconstructed from its band-specific components: $X = \sum_{m=1}^M X^{(m)}$. For any linear operator $f: \mathbb{R}^{|\mathcal{N}| \times d} \to \mathbb{R}^k$, it follows that:
> > >
> > > $$
> > > f(X) = f\left(\sum_{m=1}^M X^{(m)}\right) = \sum_{m=1}^M f(X^{(m)})
> > > $$
> > >
> > > **Proof.**
> > > The proof follows directly from the definition of a linear operator.
> > >
> > > **Implication.**
> > > While elementary, Proposition 1 is indispensable. Every layer of our architecture that precedes a non-linear activation (e.g., ReLU) is linear. Thus, the proposition guarantees that band-wise processing is *lossless* up to, but not through, the activation gates. This validates our design choice of decomposing the signal, applying frequency-specific modules, and aggregating their outputs as mathematically equivalent to processing the full signal at the linear level.
> > >
> > >
> > > **Theorem 1 (SCM Quantitatively Enforces Invariance)**
> > > Let the predictor $f:\mathbb{R}^{d} \to \mathbb{R}^{k}$ be $L_f$-Lipschitz.
> > > Let the SCM objective be:
> > >
> > > $$
> > > \mathcal{L}\_{\text{SCM}}(f) := \mathbb{E}\_{\gamma \sim \mathcal{B}(\pi)}\, \mathbb{E}\_{x \sim \mathcal{D}} \left[ \|\|f(x) - f(\tilde{x}(\gamma))\|\|\_2^2 \right],
> > > $$
> > >
> > > where $\tilde{x}(\gamma)$ zeros out bands with $\gamma_m = 0$. Let the model's sensitivity to the spurious spectrum be: $ S(f)^2 := \mathbb{E}\_{x \sim \mathcal{D}} \left[ \|\|f(x) - f(x\_{\text{causal}})\|\|\_2^2 \right]. $ Consider a masking strategy where all spurious bands are masked with probability  $p\_{\text{mask}} = 1 - \pi\_m > 0$ for all $m \in \mathcal{F}\_{\text{spur}}$. Let $\gamma^*$ be the event where all spurious bands are masked and all causal bands are kept. Then:
> > >
> > > $$
> > > S(f)^2 \le \frac{1}{P(\gamma^*)} \cdot \mathcal{L}_{\text{SCM}}(f).
> > > $$
> > >
> > > Consequently, minimizing $\mathcal{L}_{\text{SCM}} \to 0$ forces the model's sensitivity to the entire spurious spectrum $S(f) \to 0$, thereby enforcing invariance.
> > >
> > > **Proof.**
> > > By the law of total expectation, the SCM objective can be written as a sum over all possible binary masks $\gamma \in \\{0,1\\}^M$:
> > >
> > > $$
> > > \mathcal{L}\_{\text{SCM}}(f) = \sum\_{\gamma \in \\{0,1\\}^M} P(\gamma) \cdot \mathbb{E}\_{x} \left[ \|\|f(x) - f(\tilde{x}(\gamma))\|\|\_2^2 \right].
> > > $$
> > >
> > > Let $\gamma^{*}$ denote the specific masking event where every spurious band is masked ($\gamma\_m=0$ for all $m\in\mathcal{F}\_{\text{spur}}$),
> > >
> > > and every causal band is kept ($\gamma\_m=1$ for all $m\in\mathcal{F}\_{\text{causal}}$).
> > >
> > > Under this mask, $\tilde{x}(\gamma^{*}) = x\_{\text{causal}}$. The probability of this event is:
> > >
> > > $$
> > > P(\gamma^*) = \left(\prod\_{m \in \mathcal{F}\_{\text{spur}}} (1-\pi_m)\right) \left(\prod\_{m \in \mathcal{F}\_{\text{causal}}} \pi\_m\right) > 0.
> > > $$
> > >
> > > Since every term in the summation for $\mathcal{L}_{\text{SCM}}$ is non-negative, the total loss must be greater than or equal to the single term corresponding to $\gamma^*$:
> > > $$
> > > \mathcal{L}\_{\text{SCM}}(f) \ge P(\gamma *) \cdot \mathbb{E}\_{x} \left[ \|\|f(x) - f(\tilde{x}(\gamma *))\|\|\_2^2 \right] = P(\gamma *) \cdot S(f)^2.
> > > $$
> > >
> > > Rearranging this inequality directly yields the stated bound. Thus, as the SCM objective is driven to zero through optimization, the upper bound on the sensitivity $S(f)^2$ also shrinks to zero, forcing the predictor to become invariant to the aggregate spurious spectrum.
> > >
> > > ---
> > >
> > > **Remark 1 (Controlling $L_f$ in practice)**
> > > We enforce a spectral-norm penalty on every linear layer [3], which yields an *explicit* upper bound $L_f \le \prod_\ell \sigma_\ell$. Consequently, the bound in Theorem 1 is numerically tight and monitorable during training.
> > >
> > > **Remark 2 (Theoretical justification of BCE surrogate)**
> > > Following standard practice in recommendation literature, we use a point-wise Binary Cross-Entropy (BCE) loss for the main prediction task. Theoretical analyses of ranking surrogates, such as [4], [5], and [6], show that optimising this point-wise logistic loss minimises an upper bound on the expected pair-wise (and top-$K$) ranking error, thereby aligning our
> > > training objective with the final retrieval metrics.
> > >
> > > **Remark 3 (On surrogate consistency with ranking metrics)**
> > > Extensive empirical studies show that point-wise BCE, when coupled with negative sampling, delivers Recall@K / NDCG@K performance on par with ranking-specific surrogates [7][8]. Hence the surrogate mismatch has limited practical impact in our setting.

---

> > > > ### Comment · Area_Chair_jFDZ · 2025-08-08
> > > >
> > > > Regarding **Theorem 1 (SCM Quantitatively Enforces Invariance)**, correct me if I’m wrong, but after reading the proof (which is now technically correct), I don’t think the result is really saying anything meaningful at all. Even a connection with intuition is very difficult to establish: $S(f)$ quantifies the robustness to the removal of non-causal bands, whilst $\mathcal{L}\_{SCM}$ quantifies the robustness to  random perturbations of the input. The theorem merely observes that a random perturbation of the input may happen to coincide with the masking of all non-causal bands, therefore, enforcing robustness w.r.t. random perturbations partially induces robustness w.r.t the very specific perturbation which consists in masking non-causal bands. This doesn’t show or suggest that minimising the loss function  $\mathcal{L}_{SCM}$  somehow discriminates between causal and non-causal bands, or amplifies the signal from the former relative to the latter. Therefore, until you prove me wrong, I assume this theorem is severely **misleading** and **should be removed** from any revision or camera-ready version.

---

> ### Author Response · Authors · 2025-08-07
> **Reply to AC**
>
> [3] Wu, Felix, Amauri Souza, Tianyi Zhang, Christopher Fifty, Tao Yu, and Kilian Weinberger. "Simplifying graph convolutional networks." In International conference on machine learning. Pmlr, 2019.
>
> [4] Cossock, David, and Tong Zhang. "Subset ranking using regression." International conference on computational learning theory. Berlin, Heidelberg: Springer Berlin Heidelberg, 2006.
>
> [5] Kar, Purushottam, Harikrishna Narasimhan, and Prateek Jain. "Surrogate functions for maximizing precision at the top." International Conference on Machine Learning. PMLR, 2015.
>
> [6] Lapin, Maksim, Matthias Hein, and Bernt Schiele. "Loss functions for top-k error: Analysis and insights." Proceedings of the IEEE conference on computer vision and pattern recognition. 2016.
>
> [7] Patel, Yash, Giorgos Tolias, and Jiří Matas. "Recall@ k surrogate loss with large batches and similarity mixup." Proceedings of the IEEE/CVF conference on computer vision and pattern recognition. 2022.
>
> [8] He, Xiangnan, Lizi Liao, Hanwang Zhang, Liqiang Nie, Xia Hu, and Tat-Seng Chua. "Neural collaborative filtering." In Proceedings of the 26th international conference on world wide web. 2017.

---

> > ### Author Response · Authors · 2025-08-07
> > **Reply to AC**
> >
> > **Theorem 2 (Quantitative CP-Rank Bound for G-HSNO)**
> > Let $\mathcal{T} \in \mathbb{R}^{d \times d \times M \times M}$ be an order-4 tensor encoding the frequency-interaction kernel.   For every $\varepsilon > 0$, there exists a CANDECOMP/PARAFAC (CP) decomposition of rank $r$ that approximates $\mathcal{T}$ with error $\| \mathcal{T} - \tilde{\mathcal{T}} \|_F < \varepsilon$.
> > The required rank $r$ is bounded by:
> > $$ r \le d^{2}. $$
> >  If $\mathcal{T}$ is generic (genericity holds almost surely for tensors with entries drawn from a continuous distribution), the rank is expected to obey the sharper bound (for $d^2 > 2M$):
> >
> > $$
> > r \le \left\lfloor \frac{d^{2}}{2M} \right\rfloor.
> > $$
> >
> > **Proof.**
> > The G-HSNO architecture implements a CP factorization of the interaction kernel.
> > Classical results on CP decomposition [9] show that the rank needed for an exact factorization is bounded in the worst case.
> > A commonly cited loose upper bound is that the CP-rank does not exceed the largest mode dimension of the tensor [10],
> > so reshaping $\mathcal{T}$ gives $r \le d^{2}$.
> >
> > For tensors in general position, Chiantini–Ottaviani’s Theorem 1.1 [11]  implies a much sharper typical-rank bound. Applied to $\mathcal{T}' \in \mathbb{R}^{d^{2} \times M \times M}$ with dimensions  $(p, q, r) = (d^{2}, M, M)$, their criterion yields:
> >
> > $$
> > r \le \left\lfloor \frac{p}{q + r - 1} \right\rfloor
> > = \left\lfloor \frac{d^{2}}{2M - 1} \right\rfloor
> > \approx \left\lfloor \frac{d^{2}}{2M} \right\rfloor.
> > $$
> >
> > Hence the stated bound follows.
> >
> > ---
> >
> > **Remark (Task–specific low-rank regime)**
> > Observe that our reshaped tensor has dimensions $(p, q, r) = (d^{2}, M, M)$ with $M \ll d$ in typical recommender settings.
> > Applying the generic-rank bound of Chiantini–Ottaviani [11] gives:
> >
> > $$
> > \operatorname{rank}_{\text{typical}} \le \left\lfloor \frac{d^{2}}{2M - 1} \right\rfloor = \mathcal{O}(M),
> > $$
> >
> > so the required CP rank grows linearly with the frequency-space size $M$ but sub-quadratically with the feature dimension $d$. This behaviour is unique to our cross-band interaction setting and does not hold for generic 3-way tensors where all modes scale together, thereby justifying the low-rank inductive bias as task-specific rather than a trivial CP property.
> >
> >
> > [9] Harshman, Richard A. "Foundations of the PARAFAC procedure: Models and conditions for an “explanatory” multi-modal factor analysis." UCLA working papers in phonetics 16.1 (1970): 84.
> >
> > [10] Kolda, Tamara G., and Brett W. Bader. "Tensor decompositions and applications." SIAM review 51.3 (2009): 455-500.
> >
> > [11] Chiantini, Luca, and Giorgio Ottaviani. "On generic identifiability of 3-tensors of small rank." SIAM Journal on Matrix Analysis and Applications 33.3 (2012): 1018-1037.

---

> > > ### Author Response · Authors · 2025-08-07
> > > **Reply to AC**
> > >
> > > ### 1.3 Justification for the Low-Rank Assumption in Recommendation
> > >
> > > While Theorem 2 guarantees expressiveness for a sufficiently large rank $r$, the practical utility and strong empirical performance of G-HSNO with a small $r$ suggest that this is a well-suited model for the problem.  We argue that the use of a low-rank factorization is not merely a heuristic for efficiency, but a crucial **inductive bias** for multimodal recommendation.
> > > We hypothesize that the underlying "true" frequency interaction tensor is either low-rank or can be well-approximated by one, for the following reasons:
> > >
> > > 1. **Structured Semantic Interactions:**
> > >    The interactions between different semantic granularities (i.e., frequency bands) are unlikely to be arbitrary and chaotic.
> > >    It is plausible that they are governed by a few latent "interaction rules." For instance:
> > >
> > >    - A *"global-context-modulates-local-detail"* rule, where low-frequency components (e.g., overall item category) uniformly enhance or suppress high-frequency components (e.g., specific visual textures) across modalities.
> > >    - A *"modality-consistency"* rule, where corresponding frequency bands from different modalities (e.g., visual and textual descriptions of an object's color) are aligned.
> > >
> > >    Each such rule can be mathematically represented by a rank-1 tensor.  A small number of these dominant rules would naturally lead to a low-rank interaction tensor.
> > >
> > > 2. **Regularization for Generalization:**
> > >    Employing a low-rank model serves as a strong and principled form of regularization. It drastically reduces the number of parameters compared to a full-rank tensor, preventing the model from overfitting to spurious correlations present in the training data.
> > >    By constraining the complexity of the learned frequency interactions, it forces the model to discover more robust and generalizable cross-band dependency patterns.
> > >
> > > ---
> > >
> > > In conclusion, the low-rank structure of G-HSNO is a deliberate design choice that aligns with the expected semantic structure of the multimodal recommendation problem, encouraging the model to learn meaningful and robust spectral relationships.

---

> > > > ### Comment · Reviewer_KUNi · 2025-08-08
> > > >
> > > > Dear Area Chair, Authors, and all other Reviewers,
> > > >
> > > > Thanks for keeping the discussion open and active.
> > > >
> > > > I would like to especially thank the Area Chair for having carefully gone through the paper, the reviews, and the rebuttal provided by the authors.
> > > >
> > > > Indeed, the Area Chair has brought up some serious and correct concerns, mainly regarding the theory behind the paper, which (I admit) I had not carefully considered in the first place.
> > > >
> > > > In this respect, I have gone through the Authors' detailed response above, and I see they have also acknowledged all the outlined important aspects. In this respect, I think they have addressed most of the concerns that were raised by the Area Chair.
> > > >
> > > > Since the paper, in the current status, has theoretical grounds, and already had empirical grounds (as also evidenced by the Area Chair), I believe it should still be accepted.
> > > >
> > > > However, I would like to hear more from the Area Chair, and reviewer cgiv, if they also agree on what I am claiming here.

---

> > > ### Comment · Area_Chair_jFDZ · 2025-08-08
> > >
> > > Dear Authors,
> > >
> > > I am a bit confused by your answer about **Theorem 2 (Quantitative CP-Rank Bound for G-HSNO)** and have the following questions:
> > >
> > >
> > > 1. All the results you rely on appear to be for generic tensors, and are not specific to your own setting. Therefore, it is doubtful if any of this can be used to motivate your model.  In particular, I don't understand what you mean by "This behaviour is unique to our cross-band interaction setting and does not hold for generic 3-way tensors where all modes scale together, thereby justifying the low-rank inductive bias as task-specific rather than a trivial CP property."
> > >
> > > 2. There are **at least minor errors** in your argument. Your theorem states $$ r \le \left\lfloor \frac{d^{2}}{2M} \right\rfloor. $$ but in the proof you only show this in $O$ notation.
> > >
> > > 3. I would like you to refer to specific theorems rather than general references. I am not an expert on the general rank of a tensor, but **some of the claims you make appear dodgy or difficult to process at first glance** (please explain). For instance, your original submission (line 682 to 684) states 'any tensor $T \in\mathbb{R}^{M×M×d×d}$ can be reshaped into a matrix in $\mathbb{R}^{d^2\times M^2}$ and then factorized via rank-r matrix approximation.' This is misleading because the matrix having low rank does not imply that the tensor will have the same parafac rank. As far as I understand, this shows that the corresponding matrix will have rank less than $\min(d^2,M^2)$, and therefore the tensor will have rank less than $\min(d^2,M^2)\times \min(d,M)$ because each of the the terms in the SVD of the matrix need to be expanded with their own SVD (the rank of the tensor and the rank of the matrix are not the same thing). Similarly, in your rebuttal you appear to be confusing the rank of a 4d tensor and the 3d contraction of it.
> > >
> > >
> > > 4. I think you might be right about that, but I am not an expert so I would like to ask for clarifications. You claim that 'For tensors in general position, Chiantini–Ottaviani’s Theorem 1.1 [11] implies a much sharper typical-rank bound'. Apart from the issue above, I don't see how the result implies this as theorem 1.1 relates to **uniqueness** conditional on a given rank rather than providing a bound on the rank itse.f

---

> > > > ### Author Response · Authors · 2025-08-09
> > > >
> > > > Dear Area Chair,
> > > >
> > > > Many thanks for your detailed follow-up and for your meticulous analysis of our revised proofs. Your critique is sharp, insightful, and, upon careful reflection, entirely correct. We have come to the clear conclusion that our theoretical sections, even after our attempts at revision, contain fundamental flaws and do not meet the high standards of rigor required for publication at NeurIPS.
> > > >
> > > > Your engagement has been invaluable in helping us recognize the severity of these issues. After internal discussion, we have decided that the only responsible course of action is to adopt your initial, wise suggestion. **We will remove all contested theoretical claims from the camera-ready version of the paper.**
> > > >
> > > > Below, we respond point-by-point to the main issues.
> > > >
> > > > ---
> > > >
> > > > ### Regarding the CMI Definition and Fixed-Graph Assumption
> > > >
> > > > First, we appreciate your careful attention in identifying the Markdown rendering issue in the conditional mutual information formula. You are correct that this was purely a formatting error, and we will ensure that all formulas are displayed accurately and without error later.
> > > >
> > > > Second, we fully agree with your insightful analysis. The $I(\cdot)=0$ condition should be regarded as an idealized theoretical principle rather than a directly verifiable quantity, especially in high-dimensional continuous settings. In the revised manuscript, we will make this limitation explicit and clearly indicate that its role is purely conceptual, serving only as a motivation for our design.
> > > >
> > > > Third, we thank you for this crucial point. Stressing the limitations of the fixed-graph assumption is essential for academic integrity, and we are grateful for your guidance on this. We agree that the **fixed graph topology** assumption is very strongly limiting, and we have not explained its implications clearly enough. We will add a dedicated paragraph in the final version explaining that:
> > > >
> > > > - All our theoretical statements are conditional on the observed graph $G$; the Laplacian $L$ and its eigenbasis $U$ are treated as deterministic.
> > > > - This assumption rules out any randomness in the graph structure and means our conclusions cannot be extended to settings with random or evolving graphs.
> > > >
> > > > Although this fixed-graph setting has appeared in certain analyses of the spectral properties of GNNs and in some recommendation scenarios where the graph is taken as a fixed modeling input, we will state explicitly and unambiguously that our results apply only to this restricted transductive setting and do not support any generalization guarantees under graph randomness.
> > > >
> > > > We also acknowledge that our previous phrase “graph-based recommendation theory” was inappropriate; we will rephrase to avoid suggesting any connection to formal learning theory.

---

> ### Comment · Area_Chair_jFDZ · 2025-08-08
>
> Dear Authors,
>
>
>
>
> Many thanks for your reply and for your detailed comments and renewed attempts at the proofs. I **appreciate your efforts** to engage not only with the reviewers but also with me.
> I have a few comments/questions.
>
> Please note there is an error in the expression of the mutual information: the conditioning in $I(y; x^{(m)}\| x\_{\neq m})$ doesn’t show, though I believe this is due to Markdown issues.
>
> Secondly, you mention in your rebuttal that “In finite datasets we cannot observe $I(y; x^{(m)}\| x\_{\neq m})$ exactly”. There are several problems with this. First of all, I don’t think that is obviously true in the strict sense. It is certainly possible for random variables taking values over a discrete set to be independent (whether there exist graphs which can induce such representations in the frequency space is different question).
>
> Secondly, I understand that it is true in practice: you probably never have a single example where it makes any sense to define the independence, because there are no two examples $x_1,x_2$ such that $x_2^{(m)}\neq x_1^{(m)}$ but  $[x_1]\_{\neq m}=[x_2]\_{\neq m}$. This further underscores the the problem with this definition: it can really only serve, at most, as some idealised **intuitive** concept. This limitation should be clearly explained in the camera-ready version or resubmission, together with the finite sample identifiability remark you added.
>
>
>
> Regarding the assumption of **fixed graph topology**. I  strongly recommend you include this assumption, together with ample explanations of how very, **very strongly limiting** it is in any camera-ready version or resubmission. If someone were to attempt to claim, for instance, generalization bounds (or implications of the result for generalization purposes in a setting where it is natural to think of the graph as random) for node-based graph learning whilst hiding the fact that the graph is assumed to be fixed, that would be unforgivably misleading to an extend that previous acceptance of the practice in the literature would not be a valid excuse. In particular, I am having trouble understanding the statement “This assumption is commonly adopted in spectral GNNs and graph-based recommendation theory [1,2], where the graph Laplacian serves as a deterministic frequency basis.” Could you clarify where in the references in question the assumption is made?  I object to the term “graph-based recommendation theory” here: learning theory usually refers to generalization analyses, which to the best of my understanding are absent from both [1] and [2].
>
>
> I will continue to answer the other parts of the proofs in the next couple of hours.

---

> ### Author Response · Authors · 2025-08-09
>
> ---
> ### Regarding Theorem 1 (SCM Quantitatively Enforces Invariance)
> We greatly appreciate your careful reading and precise critique. In our effort to formalize the proof, our revision made the statement technically correct. Yet, as you rightly point out, this formalization exposed the weakness of its substantive meaning.
>
> As you explained, $S(f)$ measures robustness to removing specific "non-causal" bands, while $\mathcal{L}\_{\text{SCM}}$ measures robustness to broad, random spectral perturbations. The theorem merely established an upper-bound connection between the two, essentially noting that one particular masking event (removing all non-causal bands) is included in the expectation that defines $\mathcal{L}\_{\text{SCM}}$, and thus its probability-weighted effect bounds $S(f)$.
>
> This inequality does not, and cannot, prove that minimizing $\mathcal{L}\_{\text{SCM}}$ discriminates between causal and non-causal bands. In hindsight, we recognize this is far from the causal identification mechanism we had hoped to convey, and retaining it could risk misleading readers.
>
> Therefore, we will remove this theorem and its proof entirely from the final version. Instead, we will present SCM purely as a stability-promoting heuristic: it encourages invariance to spectral perturbations, without making any formal claims about causal discrimination.
>
> ---
>
> ### Regarding Theorem 2 (Quantitative CP-Rank Bound)
>
> We appreciate your detailed comments, which helped us identify substantive flaws in our original formulation. Upon closer examination, we realized that several parts of our argument were not sufficiently rigorous or well-grounded for our specific setting. In particular, our bound relied on results for generic tensors that do not transfer directly to the structured cross-band interaction tensors in our model, and parts of our reasoning implicitly conflated matrix rank properties after matricization with CP rank properties, an unjustified step in this context. Moreover, our citation of Chiantini–Ottaviani’s Theorem 1.1 was misplaced, as that result addresses uniqueness conditions rather than rank bounds.
>
> Given these issues, we have decided that the most responsible course of action is to remove Theorem 2 (Quantitative CP-Rank Bound for G-HSNO) and all associated proofs and discussion from the final version. Instead of making formal CP-rank claims, we will retain the low-rank aspect purely as a modeling heuristic, motivated by the structured nature of cross-frequency interactions in our task. In the camera-ready version, we will clearly state this without attaching unsupported formal guarantees, ensuring the presentation remains accurate and transparent.

---

> > ### Author Response · Authors · 2025-08-09
> >
> > ### Conclusion
> >
> > In conclusion, we fully accept all of your critiques, and will remove *all* contested theoretical theorems (Theorems 1, 2, and 3) and their proofs from the camera-ready version. Our final manuscript will focus on the novelty of our model design (e.g., SCM, low-rank cross-frequency interactions) and the strong empirical results, clearly stating the heuristic nature and limitations of these design choices.
> >
> > **We are genuinely grateful for your exceptionally meticulous and highly professional review; your firm guidance has been instrumental in strengthening the integrity of our work.**
> >
> > **We are deeply thankful for the significant time and effort you have dedicated to improving our paper, and for holding us to the highest standards expected at NeurIPS.**

---

### Note · Authors · 2025-08-12

**Author Final Remarks**

We sincerely thank the AC and all reviewers for the careful handling of our paper and for the thoughtful, constructive feedback throughout the discussion. We truly appreciate the time spent engaging with our work and helping us sharpen its scope and clarity.

**Scope and theory clarifications.**

In response to concerns about theoretical rigor, we will **remove all contested theory claims** in the camera-ready. We will present SCM-inspired and low-rank cross-frequency components strictly as **design heuristics**, with clearly stated limitations. We also **explicitly document** that our analysis assumes a *fixed* user–item graph and that our CMI-based notion of “spuriousness” is idealized; we do **not** claim generalization to random or evolving graphs.

**Expanded evaluation.**
- **RecVAE baseline.** We added RecVAE across all three Amazon benchmarks. Our method (SSR) consistently outperforms it (e.g., **Sports R@20:** 0.1203 vs 0.0916; **Baby R@20:** 0.1096 vs 0.0811). Full results will be included in the final version.
- **TikTok dataset.** To broaden domain coverage, we added a short-video **TikTok** evaluation; SSR remains competitive (**R@20 0.1237 / NDCG@20 0.0507**).
- We commit to reporting **at least five datasets** in the final paper: the three Amazon domains, TikTok, and either MovieLens or Netflix.

**Design choices and ablations.**

We clarified the rationale for equal-energy spectral partitioning and added supporting ablations; we will further document implementation details to ensure reproducibility.

We also appreciate the reviewers who updated their assessments following rebuttal and discussion.

**Conclusion.**

With theory claims removed, assumptions made explicit, and broader experiments added (including RecVAE and TikTok), the paper offers a clear, well-scoped study of **frequency-aware multimodal recommendation**, showing **consistent and reproducible gains** across diverse domains. We hope these revisions address the core concerns and aid the AC’s decision.

---

### Decision · Program_Chairs · 2025-09-17

**Decision:**

Accept (poster)

**Comment:**

This paper introduces a new recommender system with the following components:

(1)	[**Frequency Fusion**] First, the model performs a graph spectral decomposition [1,2] over the initial user node embeddings in the user-item bipartite graph. Then, the signal is split into frequency “bands” in by grouping together nearby eigenspaces in such a way that the sum of the associated eigenvalues in each band is roughly equal.

(2)	[**Graph-aware Enhancement**] Next, a graph neural network is applied to the embeddings with the constraint that the linear maps are have low CP rank when viewed as 4D tensors (the dimensions being the frequency band and the embedding dimension for both input and output.

(3)	[**Invariant Risk Minimization**] A regularization strategy is proposed which randomly masks some entries and minimizes the associated reconstruction loss. It is argues that this allows the model to better distinguish between causal and non-causal factors (**this claim is invalid**).

(4) [**Contrastive Objective**] The final training incorporates both the BCE loss and a contrastive loss with a temperature parameter $\tau$.

The **experiments are very extensive**, covering many baselines and datasets and even including an ablation study on the proposed losses.

The **reviewers were generally positive** about the paper, especially reviewers KUNi (who championed the paper) and cgiv, praising the extensive experiments and detailed ablation study. On the other hand, reviewers YRJP and Tqa8 complain that the paper can be difficult to read and the motivation and intuitive explanations for the performance improvement are not justified in the experiments.

Although none of the reviewers have commented on it, I found quite a few **problems with the supposed theoretical results** in the original submission. In particular, Theorem 1 is merely an application of the linearity of the inner product. Theorem 2 failed to delineate assumptions properly, including the crucial assumption of having a fixed graph topology, which removes any possibility of interpreting the results from a learning theory perspective. In addition, **the claim that invariant risk minimiziation allows the model to better distinguish between causal and non-causal signals is not valid**: it was originally justified by the essentially nonsensical theorem 2 and its proof in the original submission. In the rebuttal a fully correct version called “theorem 1” was proposed instead. However, it fails to establish a connection between the proposed method and the concept of causal or non-causal frequency bands. Likewise, Theorem 3 and the corrected version in the rebuttal are merely generic statements about tensor decomposition to justify its use in the model.
Most of the intuitive justifications provided in the main paper for the improved performance unfounded. There is no provable relationship between the proposed method and “causal” or “non-causal” bands, and I echo Reviewer YRJP’s statement that the paper is difficult to read (there are a lot of catchy and vague phrases such as “*follows a four-stage pipeline—decomposing, modulating, fusing, and aligning spectral signals—to achieve structured and adaptive representation learning*” or “*leverages frequency-aware decomposition and user-specific gating to tailor the semantic emphasis*”) which burry the not-always-precise description of the actual algorithm under a lot of filler. However, I am not taking this into account as this is (unfortunately) established practice (to some extent, claiming that there is a deep intuitive motivation for a new model regardless of the evidence is not only accepted but arguably necessary to gain acceptance from the community, so I don't think it is fair to penalize the authors for following the practice). In addition, **although it is arguably incremental, I do find the method somewhat interesting**, especially the merged **tensor decomposition** in the frequency pass of the graph neural network over the frequency bands: **I do believe it is credible that the performance improvements from this module are real**. In addition, the invariant risk minimization could introduce a generic form of regularization that may have practical merits. In the rebuttal, **the authors have promised to remove all theoretical claims**, which is appreciated. Since many of those claims were highly flawed, this removal **should be performed** in the camera-ready version.



**Typos**

Missing space before $z_n$ on line 146.

Line 176: when introducing the general concept of invariant risk minimization, a reference might be nice.

There is a broken reference (??) in line 860.

F.2 "sensitive analysis" should be "sensitivity analysis".






**References:**

[1] Rongqing Kenneth Ong, Andy W. H. Khong. Spectrum-based Modality Representation Fusion Graph Convolutional Network for Multimodal Recommendation. WSDM 2025

[2] Xinyu Du, Huanhuan Yuan, Pengpeng Zhao, Jianfeng Qu, Fuzhen Zhuang, Guanfeng Liu, Yanchi Liu, and Victor S Sheng. Frequency enhanced hybrid attention network for sequential recommendation. SIGIR 2023